# Mitigating Gradient Bias in Multi-objective Learning: A Provably Convergent Approach

**Heshan Fernando**[1], **Han Shen**[1], **Miao Liu**[2], **Subhajit Chaudhury**[2], **Keerthiram Murugesan**[2], **Tianyi Chen**[1]

[1]Rensselaer Polytechnic Institute  [2]IBM Thomas J. Watson Research Center

## Abstract

Machine learning problems with multiple objectives appear either i) in learning with multiple criteria where learning has to make a trade-off between multiple performance metrics such as fairness, safety and accuracy; or, ii) in multi-task learning where multiple tasks are optimized jointly, sharing inductive bias among them. These multiple-objective learning problems are often tackled by the multi-objective optimization framework. However, existing stochastic multi-objective gradient methods and their recent variants (e.g., MGDA, PCGrad, CAGrad, etc.) all adopt a biased gradient direction, which leads to degraded empirical performance. To this end, we develop a stochastic multi-objective gradient correction (MoCo) method for multi-objective optimization. The unique feature of our method is that it can guarantee convergence without increasing the batch size even in the nonconvex setting. Simulations on supervised and reinforcement learning demonstrate the effectiveness of our method relative to state-of-the-art methods.

## 1 Introduction

Multi-objective optimization (MOO) involves optimizing multiple, potentially conflicting objectives simultaneously. Recently, MOO has gained attention in various application settings such as optimizing hydrocarbon production (You et al., 2020), tissue engineering (Shi et al., 2019), safe reinforcement learning (Thomas et al., 2021), and training neural networks for multiple tasks (Sener & Koltun, 2018). We consider the stochastic MOO problem as

$$\min_{x \in \mathcal{X}} F(x) := (\mathbb{E}_\xi[f_1(x, \xi)], \mathbb{E}_\xi[f_2(x, \xi)], \ldots, \mathbb{E}_\xi[f_M(x, \xi)]) \tag{1}$$

where $\mathcal{X} \subseteq \mathbb{R}^d$ is the feasible set, and $f_m : \mathcal{X} \mapsto \mathbb{R}$ with $f_m(x) := \mathbb{E}_\xi[f_m(x, \xi)]$ for $m \in [M]$. Here we denote $[M] := \{1, 2, \ldots, M\}$ and denote $\xi$ as a random variable. In this setting, we are interested in optimizing all of the objective functions simultaneously without sacrificing any individual objective. Since we cannot always hope to find a common variable $x$ that achieves optima for all functions simultaneously, a natural solution instead is to find the so-termed *Pareto stationary point* $x$ that cannot be further improved for all objectives without sacrificing some objectives. In this context, a multiple gradient descent algorithm (MGDA) has been developed for achieving this goal (Désidéri, 2012). The idea of MGDA is to iteratively update the variable $x$ via a common descent direction for all the objectives through a time-varying convex combination of gradients from individual objectives. Recently, various MGDA-based MOO algorithms have been proposed, especially for multi-task learning (MTL) (Sener & Koltun, 2018; Chen et al., 2018; Yu et al., 2020a; Liu et al., 2021a).

While the deterministic MGDA algorithm and its variants are well understood in literature, only little theoretical study has been taken on its stochastic counterpart. Recently, (Liu & Vicente, 2021) has introduced the stochastic multi-gradient (SMG) method as a stochastic counterpart of MGDA (see Section 2.3 for details). To establish convergence, however, (Liu & Vicente, 2021) requires a strong assumption on the fast decaying first moment of the gradient, which was enforced by linearly growing the batch size. While this allows for analysis of multi-objective optimization in stochastic setting, this may not be true for many MTL tasks in practice. Furthermore, the analysis in (Liu & Vicente,

---

The work was supported by the National Science Foundation CAREER project 2047177 and the RPI-IBM Artificial Intelligence Research Collaboration. Correspondence to: Tianyi Chen (chentianyi19@gmail.com).

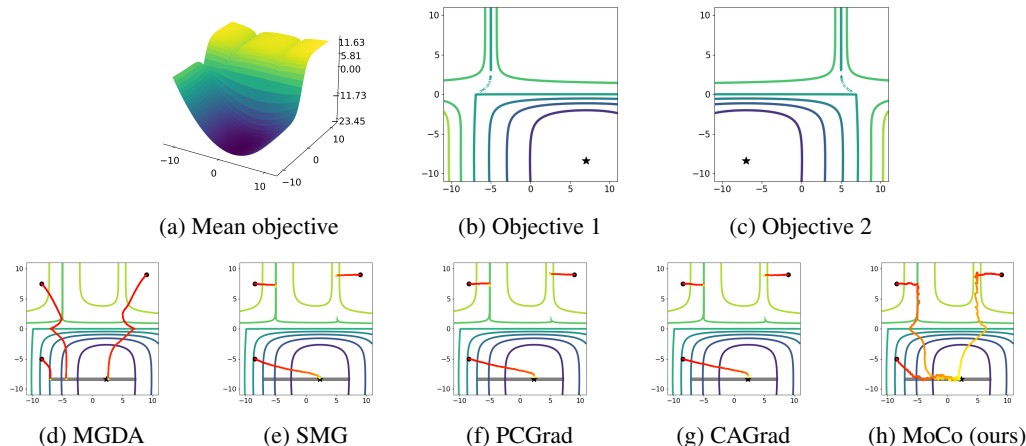

Figure 1: A toy example from (Liu et al., 2021a) with two objective (Figures 1b and 1c) to show the impact of gradient bias. We use the mean objective as a reference when plotting the trajectories corresponding to each initialization (3 initializations in total). The starting points of the trajectories are denoted by a **black** •, and the trajectories are shown fading from **red** (start) to **yellow** (end). The Pareto front is given by the **gray** bar, and the **black** ⋆ denotes the point in the Pareto front corresponding to equal weights to each objective. We implement recent MOO algorithms such as SMG (Liu & Vicente, 2021), PCGrad (Yu et al., 2020a), and CAGrad (Liu et al., 2021a), and MGDA (Désidéri, 2012) alongside our method. Except for MGDA (Figure 1d) all the other algorithms only have access to gradients of each objective with added zero mean Gaussian noise. It can be observed that SMG, CAGrad, and PCGrad fail to find the Pareto front in some initializations.

2021) cannot cover the important setting with non-convex multiple objectives, which is prevalent in challenging MTL tasks. This leads us to a natural question:

*Can we design a stochastic MOO algorithm that provably converges to a Pareto stationary point without growing batch size and also in the nonconvex setting?*

**Our contributions.** In this paper, we answer this question affirmatively by providing the first stochastic MOO algorithm that provably converges to a Pareto stationary point without growing batch size. Specifically, we make the following major contributions:

- **C1)** **(Asymptotically unbiased multi-gradient).** We introduce a new method for MOO that we call the stochastic Multi-objective gradient with Correction (MoCo) method. MoCo is a simple algorithm that addresses the convergence issues of stochastic MGDA and provably converges to a Pareto stationary point under several stochastic MOO settings. We use a toy example in Figure 1 to demonstrate the empirical benefit of our method. In this example, MoCo is able to reach the Pareto front from all initializations, while other MOO algorithms such as SMG, CAGrad, and PCGrad fail to find the Pareto front due to using biased multi-gradient.

- **C2)** **(Unified non-asymptotic analysis).** We generalize our MoCo method to the case where the individual objective function has a nested structure and thus obtaining unbiased stochastic gradients is costly. We provide a unified convergence analysis of the nested MoCo algorithm in smooth non-convex and convex stochastic MOO settings. To our best knowledge, this is the first analysis of smooth non-convex stochastic gradient-based MOO.

- **C3)** **(Experiments on MTL applications).** We provide an empirical evaluation of our method with existing state-of-the-art MTL algorithms in supervised learning and reinforcement learning (RL) settings, and show that our method can outperform prior methods such as stochastic MGDA, PCGrad, CAGrad, and GradDrop.

## 2 BACKGROUND

In this section, we introduce the concepts of Pareto optimality and Pareto stationarity and then discuss MGDA and its existing stochastic counterpart. We then motivate our proposed method by elaborating the challenge in stochastic MOO. The notations used in the paper are summarized in Appendix A.

## 2.1 PARETO OPTIMALITY AND PARETO STATIONARITY

In MOO, we are interested in finding the points which can not be improved simultaneously for all the objective functions, leading to the notion of *Pareto optimality*. Consider two feasible solutions to (1) $x, x' \in \mathcal{X}$. We say that $x$ dominates $x'$ if $f_m(x) \leq f_m(x')$ for all $m \in [M]$, and $F(x) \neq F(x')$. If a point $x^* \in \mathcal{X}$ is not dominated by any $x \in \mathcal{X}$, we say $x^*$ is Pareto optimal. The collection of all the Pareto optimal points are called as the Pareto set. The collection of vector objective values $F(x^*)$ for all the Pareto optimal $x^*$ is called as the Pareto front.

Akin to the single objective case, a necessary condition for Pareto optimality is *Pareto stationarity*. If $x$ is a Pareto stationary point, then there is no common descent direction for all $f_m(x)$ at $x$. Formally, $x$ is a called a Pareto stationary point if $\text{range}(\nabla F(x)^\top) \cap (-\mathbb{R}_{++}^M) = \emptyset$ where $\nabla F(x) \in \mathbb{R}^{d \times M}$ is the Jacobian of $F(x)$, i.e. $\nabla F(x) \coloneqq (\nabla f_1(x), \nabla f_2(x), \ldots, \nabla f_M(x))$, and $\mathbb{R}_{++}^M$ is the positive orthant cone. When all $f_m(x)$ are strongly convex, a Pareto stationary point is also Pareto optimal.

## 2.2 MULTIPLE GRADIENT DESCENT ALGORITHM (MGDA)

The MGDA algorithm has been proposed in (Désidéri, 2012) that can converge to a Pareto stationary point of $F(x)$. MGDA achieves this by seeking a convex combination of individual gradients $\nabla f_m(x)$ (also known as the multi-gradient), given by $d(x) = \sum_{m=1}^M \lambda_m^*(x) \nabla f_m(x)$ where the weights $\lambda^*(x) \coloneqq (\lambda_1^*(x), ..., \lambda_M^*(x))^\top$ are found by solving the following sub-problem:

$$\lambda^*(x) = \arg\min_\lambda \|\nabla F(x) \lambda\|^2 \quad \text{s. t.} \quad \lambda \in \Delta^M \coloneqq \{\lambda \in \mathbb{R}^M \mid \mathbf{1}^\top \lambda = 1, \ \lambda \geq 0\}. \tag{2}$$

With this multi-gradient $d(x)$, the $k$th iteration of MGDA is given by

$$x_{k+1} = \Pi_\mathcal{X}\left(x_k - \alpha_k d(x_k)\right) \quad \text{with} \quad d(x_k) = \sum_{m=1}^M \lambda_m^*(x_k) \nabla f_m(x_k) \tag{3}$$

where $\alpha_k$ is the learning rate, and $\Pi_\mathcal{X}$ denotes the projection to set $\mathcal{X}$. It can be shown that MGDA optimizes all objectives simultaneously following the direction $-d(x)$ whenever $x$ is not a Pareto stationary point and will terminate once it reaches a Pareto stationary point (Fliege et al., 2019).

However, in many real world applications we either do not have access to the true gradient of functions $f_m$ or obtaining the true gradients is prohibitively expensive in terms of computation. This leads us to a possible stochastic counterpart of MGDA, which is discussed next.

## 2.3 STOCHASTIC MULTI-OBJECTIVE GRADIENT AND ITS BRITTLENESS

The stochastic counterpart of MGDA, referred to as SMG algorithm, has been studied in (Liu & Vicente, 2021). In SMG algorithm, the stochastic multi-gradient is obtained by replacing the true gradients $\nabla f_m(x)$ in (2) with their stochastic approximations $\nabla f_m(x, \xi)$, where $\nabla f_m(x) = \mathbb{E}_\xi[\nabla f_m(x, \xi)]$. Specifically, the stochastic multi-gradient is given by

$$g(x, \xi) = \sum_{m=1}^M \lambda_m^g(x, \xi) \nabla f_m(x, \xi) \quad \text{with} \quad \lambda^g(x, \xi) = \arg\min_{\lambda \in \Delta^M} \left\| \sum_{m=1}^M \lambda_m \nabla f_m(x, \xi) \right\|^2. \tag{4}$$

While this change of the subproblem facilitates use of stochastic gradients in place of deterministic gradients, it raise issues in the biasedness in the stochastic multi-gradient calculated in this method.

*The bias of SMG.* To better understand the cause of this bias, consider the case $M = 2$ of (4) for simplicity. We can rewrite the problem for solving for convex combination weights as $\arg\min_{\lambda \in [0,1]} \|\lambda \nabla f_1(x, \xi) + (1 - \lambda) \nabla f_2(x, \xi)\|^2$, which admits the closed-form solution for $\lambda$ as

$$\lambda^g(x, \xi) = \left[ \frac{(\nabla f_2(x, \xi) - \nabla f_1(x, \xi))^\top \nabla f_2(x, \xi)}{\|\nabla f_1(x, \xi) - \nabla f_2(x, \xi)\|^2} \right]_{+, \frac{1}{\top}} \tag{5}$$

where $[x]_{+, \frac{1}{\top}} = \max(\min(x, 1), 0)$. It can be seen that the solution for $\lambda$ is non-linear in $\nabla f_1(x, \xi)$ and $\nabla f_2(x, \xi)$, which suggests that $\mathbb{E}[\lambda^g(x, \xi)] \neq \lambda^*(x)$ and thus $\mathbb{E}[g(x, \xi)] \neq d(x)$.

To ensure convergence, a recent approach proposed to replace the stochastic gradient $\nabla f_m(x, \xi)$ with its mini-batch version with the batch size growing with the number of iterations (Liu & Vicente,

---

**Algorithm 1** MoCo: Stochastic Multi-objective gradient with Correction

---

1: **Input** Initial model parameter $x_0$, tracking parameters $\{y_{0,i}\}_{m=1}^M$, convex combination coefficient parameter $\lambda_0$, and their respective learning rates $\{\alpha_k\}_{k=0}^K$, and $\{\beta_k\}_{k=0}^K$, $\{\gamma_k\}_{k=0}^K$.
2: **for** $k = 0, \ldots, K-1$ **do**
3:     **for** objective $m = 1, \ldots, M$ **do**
4:         Obtain gradient estimator $h_{m,k}$       ▷ either $h_{m,k} = \nabla f_m(x_k, \xi_k)$ or $h_{m,k}$ in (13)-(14)
5:         Update $y_{k+1,m}$ following (6)
6:     **end for**
7:     Update $\lambda_{k+1}$ and $x_{k+1}$ following (9)-(10)
8: **end for**
9: **Output** $x_K$

---

2021). However, this may not be desirable in practice and often leads to sample inefficiency. In multi-objective reinforcement learning settings, this means running increasingly many number of roll-outs for policy gradient calculation, which may be infeasible. On the other hand, Yang et al. (2021) also analyzes MGDA in the stochastic, smooth, and non-convex setting, and establishes convergence. However, to overcome the bias issue in stochastic MGDA, Yang et al. (2021) assumes having access to $\lambda^*(x)$, which allows access to an unbiased estimate of the true multi-gradient $d(x)$. However, this assumption is not practical since computing $\lambda^*(x)$ requires access to true gradients $\nabla f_m(x)$, which may not be true in a stochastic setting. In contrast, in the following section we propose a method that reduces the bias in multi-gradient asymptotically and enjoys provable convergence.

## 3 STOCHASTIC MULTI-OBJECTIVE GRADIENT DESCENT WITH CORRECTION

In this section, we will first propose a new stochastic update that addresses the biased multi-gradient in MOO, extend it to the nested MOO setting, and then establish its convergence result. To achieve this, we use a momentum-like gradient estimate and a regularized version of MGDA subproblem.

### 3.1 A BASIC ALGORITHMIC FRAMEWORK

We start by discussing how to obtain $\nabla f_m(x)$ without incurring the bias issue. The key idea is to approximate true gradients of each objective using a 'tracking variable', and use these approximations in finding optimal convex combination coefficients, similar to MGDA and SMG. At each iteration $k$, assuming we have access to $h_{k,m}$ which is a stochastic estimator of $\nabla f_m(x_k)$ (e.g., $h_{k,m} = \nabla f_m(x_k, \xi_k)$). We obtain $\nabla f_m(x_k)$ by iteratively updating the 'tracking' variable $y_{k,m} \in \mathbb{R}^d$ by

$$y_{k+1,m} = \Pi_{L_m}\Big(y_{k,m} - \beta_k\big(y_{k,m} - h_{k,m}\big)\Big), \quad m = 1, 2, \cdots, M, \tag{6}$$

where $\beta_k$ is the step size and $\Pi_{L_m}$ denotes the projection to set $\{y \in \mathbb{R}^d \mid \|y\| \leq L_m\}$, and $L_m$ is the Lipschitz constant of $f_m$ on $\mathcal{X}$.

Under some assumptions on the stochastic gradients $h_{k,m}$ that will be specified in Section 3.3, we can show that for a given $x_k$, the recursion in (6) admits a unique fixed-point $y_m^*(x_k)$ that satisfies

$$y_m^*(x_k) = \mathbb{E}[h_{k,m}] = \nabla f_m(x_k). \tag{7}$$

In this subsection we will first assume that $h_{k,m}$ is an unbiased estimator of $\nabla f_m(x_k)$, and will generalize to the biased estimator in the next subsection. In this case, with only one sample needed at each iteration, the distance between $y_{m,k}$ and $\nabla f_m(x_k)$ is expected to diminish as $k$ increases.

Even with an accurate estimate of $\nabla f_m(x)$, solving (1) is still not easy since these gradients could conflict with each other. As described in Section 2.2, given $x \in \mathcal{X}$, the MGDA algorithm finds the optimal scalars, denoted as $\{\lambda_m^*(x)\}_{m=1}^M$, to scale each gradient $\nabla f_m(x)$ such that $d(x) = \sum_{m=1}^M \lambda_m^*(x)\nabla f_m(x)$, and $-d(x)$ is a common descent direction for every $f_m(x)$. For obtaining the corresponding convex combinations when we do not have access to the true gradient, we propose to use $Y_k := (y_{k,1}, ..., y_{k,M}) \in \mathbb{R}^{d \times M}$ as an approximation of $\nabla F(x_k)$. In general, the solution for (2) is not necessarily unique. We overcome this issue by adding $\ell_2$ regularization. Specifically, with $\rho > 0$ denoting the regularization constant, the new subproblem is given by

$$\lambda_\rho^*(x) = \arg\min_\lambda \ \|\nabla F(x)\lambda\|^2 + \frac{\rho}{2}\|\lambda\|^2 \ \text{ s. t. } \ \lambda \in \Delta^M := \{\lambda \in \mathbb{R}^M \mid \mathbf{1}^\top \lambda = 1, \ \lambda \geq 0\}. \tag{8}$$

**Remark 1** (On the Lipschitz continuity of $\lambda_\rho^*(x)$). *Since (2) and (4) depend on $x$, the subproblems change at each iteration. To analyze the convergence of the algorithm, it is important to quantify the change of solutions $\lambda^*(x)$ and $\lambda^g(x, \xi)$ at different $x$. One natural way is to assume the afore-mentioned solutions are Lipschitz continuous in $\nabla F(x)$; see (Liu & Vicente, 2021). However, this condition does not hold in general since $\nabla F(x)$ is not positive definite at least at Pareto stationary points, and thus the solutions to (2) and (4) are not unique. We overcome this issue by adding the regularization $\rho$ to ensure uniqueness of the solution and the Lipschitz continuity of $\lambda_\rho^*(x)$ in $x$.*

With this regularized reformulation, we find $\lambda_\rho^*(x)$ by running stochastic projected gradient descent of (8), given by

$$\lambda_{k+1} = \Pi_{\Delta^M} \left( \lambda_k - \gamma_k \left( Y_k^\top Y_k + \rho I \right) \lambda_k \right), \tag{9}$$

where $\gamma_k$ is the step size, $I \in \mathbb{R}^{M \times M}$ is the identity matrix, and operator $\Pi_{\Delta^M}$ denotes the projection to the probability simplex $\Delta^M$. With $\lambda_k$ as an approximation of $\lambda_\rho^*(x_k)$ and $Y_k$ as an approximation of $\nabla F(x_k)$, we then update $x_k$ with

$$x_{k+1} = \Pi_{\mathcal{X}}(x_k - \alpha_k Y_k \lambda_k), \tag{10}$$

where $\mathcal{X}$ is a closed convex set. We have summarized the basic MoCo algorithm in Algorithm 1.

## 3.2 GENERALIZATION TO NESTED MOO SETTING

In this section we extend MoCo to the bi-level MOO setting. Recall, in the previous section, we have introduced the gradient estimator $h_{k,m}$. In the simple case where $\nabla f_m(x, \xi)$ is obtained, setting $h_{k,m} = \nabla f_m(x_k, \xi_k)$ leads to the exact solution $y_m^*(x_k) = \nabla f_m(x_k)$. However, in some practical applications as shown in Section 5.2, $\nabla f_m(x, \xi)$ is difficult to obtain, and hence $h_{k,m}$ can be biased.

To put this on concrete ground, we first consider the following nested multi-objective problem:

$$\min_{x \in \mathcal{X}} F(x) := (\mathbb{E}_\xi[f_1(x, z_1^*(x), \xi)], \mathbb{E}_\xi[f_2(x, z_2^*(x), \xi)], \dots, \mathbb{E}_\xi[f_M(x, z_M^*(x), \xi)])$$

$$\text{s.t.} \quad z_m^*(x) := \arg\min_{z \in \mathbb{R}^d} l_m(x, z) := \mathbb{E}_\varphi[l_m(x, z, \varphi)], \quad m = 1, 2, \cdots, M \tag{11}$$

where $l_m$ is a strongly-convex function, and $\varphi$ is a random variable. For convenience, we define $f_m(x, z) := \mathbb{E}_\xi[f_m(x, z, \xi)]$ and $f_m(x) := f_m(x, z_m^*(x))$. The problem (11) is a generalization of the popular bilevel optimization framework Ghadimi & Wang (2018); Hong et al. (2020); Liu et al. (2020); Ji et al. (2021); Chen et al. (2021; 2022).

Under some conditions that will be specified later, it has been shown in (Ghadimi & Wang, 2018) that the gradient of $f_m(x)$ takes the following form:

$$\nabla f_m(x) = \nabla_x f_m(x, z_m^*(x)) - \nabla_{xz}^2 l_m(x, z_m^*(x))[\nabla_{zz}^2 l_m(x, z_m^*(x))]^{-1} \nabla_z f_m(x, z_m^*(x)) \tag{12}$$

where $\nabla_x f(x, z_m^*(x)) = \frac{\partial f(x,z)}{\partial x}|_{z=z_m^*(x)}$, $\nabla_{xz}^2 l(x, z_m^*(x)) = \frac{\partial l(x,z)}{\partial x \partial z}|_{z=z_m^*(x)}$ and likewise for $\nabla_z f(x, z_m^*(x))$ and $\nabla_{zz}^2 l(x, z_m^*(x))$. Computing the unbiased stochastic estimate of (12) requires $z_m^*(x)$, which is often costly in practice. Instead, we iteratively update $z_{k,m}$ to approach $z_m^*(x_k)$ via

$$z_{k+1,m} = z_{k,m} - \beta_k \nabla_z l_m(x_k, z_{k,m}, \varphi_k). \tag{13}$$

Then we use $z_{k,m}$ to replace $z_m^*(x_k)$ in the place of (12) to compute a biased gradient estimator as

$$h_{k,m} = \nabla_x f_m(x_k, z_{k,m}, \xi_k) - \nabla_{xz}^2 l_m(x_k, z_{k,m}, \varphi_k') H_{k,m}^{zz} \nabla_z f_m(x_k, z_{k,m}, \xi_k) \tag{14}$$

where $\varphi_k, \varphi_k'$ have the same distribution as that of $\varphi$, and $H_{k,m}^{zz}$ is a stochastic approximation of the Hessian inverse $[\nabla_{zz}^2 l_m(x_k, z_{k,m})]^{-1}$. Given $x_k$, when $z_{k,m}$ reaches the optimal solution $z_m^*(x_k)$, it follows from (12) that $\mathbb{E}[h_{k,m}] = \nabla f_m(x_k)$. We summarize the algorithm for MoCo with inexact gradient in Algorithm 2 (see Appendix D). When $z_{k,m}$ is non-optimal, we quantify the error below.

**Lemma 1.** *Define $\mathcal{F}_k$ as the $\sigma$-algebra generated by $Y_1, Y_2, ..., Y_k$. Consider the sequences generated by (6), (9), (10), (13) and (14). Under certain standard assumptions that will be specified in the supplementary, we have for any $m \in [M]$ and for any $k$ that*

$$\frac{1}{K} \sum_{k=1}^K \mathbb{E}\left[\|\mathbb{E}[h_{k,m}|\mathcal{F}_k] - \nabla f_m(x_k)\|^2\right] = \mathcal{O}\left(\frac{\alpha_K^2}{\beta_K^2}\right), \quad \mathbb{E}\left[\|h_{k,m} - \mathbb{E}[h_{k,m}|\mathcal{F}_k]\|^2|\mathcal{F}_k\right] \le \sigma_0^2 \tag{15}$$

*where $\mathbb{E}[\cdot]$ is the total expectation, $\alpha_K, \beta_K$ are the learning rates, and $\sigma_0 > 0$ is a constant.*

Lemma 1 shows that the average bias of the gradient estimator will diminish if $\alpha_k$ and $\beta_k$ are chosen properly. In addition, the variance of the estimator is also bounded by a constant. Allowing biased gradient in this manner facilitates MoCo to tackle more challenging MTL tasks as highlighted below.

**Remark 2** (Connection between nested MOO with multi-objective actor-critic). *Choosing each $f_m$ in (11) to be the infinite-horizon accumulated reward and each $l_m$ to be the critic objective function will lead to the popular actor-critic algorithm in reinforcement learning (Konda & Borkar, 1999; Wen et al., 2021). In this work, we have extended this to the multi-objective case, and conducted experiments on multi-objective soft actor critic in Appendix K.3.*

## 3.3 A UNIFIED CONVERGENCE RESULT

In this section we provide the convergence analysis for our proposed method. First, we make following assumptions on the objective functions.

**Assumption 1.** *For $m \in [M]$: $f_m(x)$ is Lipschitz continuous with modulus $L_m$ and $\nabla f_m(x)$ is Lipschitz continuous with modulus $L_{m,1}$, for any $x \in \mathcal{X}$.*

Due to the $x$ update in (10), the optimal solution for $y_{k,m}$ and $\lambda_k$ sequences are changing at each iteration, and the change scales with $\|x_{k+1} - x_k\|$. In order to guarantee the convergence of $y_{k,m}$ and $\lambda_k$, the change in optimal solution needs to be controlled. The first half of Assumption 1 ensures that $\nabla f(x)$ is uniformly bounded, that is, $\|x_{k+1} - x_k\|$ is upper bounded and thus controlled. The second half of the assumption is standard in establishing the convergence of non-convex functions (Bottou et al., 2018). Next, we make an alternative version of Assumption 1 for analysis in the convex setting.

**Assumption 2.** *Function $f_m(x)$ is convex for any $m \in [M]$ and the feasible set $\mathcal{X}$ is bounded.*

Notice when $\mathcal{X}$ is bounded, then there exists a constant $C_x$ such that $\|x - x'\| \le C_x$ for any $x, x' \in \mathcal{X}$. This assumption controls $\|x_{k+1} - x_k\|$ when the objective functions are convex. Next, to unify the analysis of the nested MOO in Section 3.2 and the basic MOO in Section 3.1, we make the following assumption on the quality of the gradient estimator $h_{k,m}$.

**Assumption 3.** *For any $m \in [M]$, there exist constants $c_m, \sigma_m$ such that $\frac{1}{K}\sum_{k=1}^{K} \mathbb{E}\|\mathbb{E}[h_{k,m}|\mathcal{F}_k] - \nabla f_m(x_k)\|^2 \le c_m \alpha_K^2/\beta_K^2$ and $\mathbb{E}[\|h_{k,m} - \mathbb{E}[h_{k,m}|\mathcal{F}_k]\|^2|\mathcal{F}_k] \le \sigma_m^2$ for any $k$.*

Assumption 3 requires the stochastic gradient $h_{k,m}$ almost unbiased and has bounded variance. Compared to (Liu & Vicente, 2021, Assumption 5.2), Assumption 3 is weaker, because i) it does not require the variance $\sigma_m^2$ to decrease in the same speed as $\alpha_k^2$; and ii) it allows bias in the stochastic gradient of each objective function. In practice, the batch size is often fixed, and thus the variance is non-decreasing, which suggests one benefit of Assumption 3 over that in (Liu & Vicente, 2021).

**Lemma 2.** *Consider the sequences generated by Algorithm 1. Assume that $K \gg M$ such that $K = \mathcal{O}(M^{10})$. Then, under Assumptions 1 and 3, or Assumptions 2 and 3, if we choose step sizes $\alpha_k = \Theta(K^{-\frac{9}{10}})$, $\beta_k = \Theta(K^{-\frac{1}{2}})$, $\gamma_k = \Theta(K^{-\frac{2}{5}})$, and $\rho = \Theta(K^{-\frac{1}{5}})$, it holds that*

$$\frac{1}{K}\sum_{k=1}^{K} \mathbb{E}\left[\|d(x_k) - Y_k\lambda_k\|^2\right] = \mathcal{O}(K^{-\frac{1}{5}}). \tag{16}$$

With suitable choice of $\rho$, as $\lambda_k$ and $Y_k$ converge to $\lambda_\rho^*(x_k)$ and $\nabla F(x_k)$ respectively, the update direction $Y_k\lambda_k$ for $x_k$ converges to $d(x_k) = \nabla F(x_k)\lambda^*(x_k)$, which is the desired MGDA direction. It can be seen that our method achieves vanishingly small expected error in stochastic multi-gradient asymptotically for all trajectories, while SMG fails to reduce the error in multi-gradient.

The following theorem then captures the convergence of $x_k$ under convex objective functions.

**Theorem 1.** *Consider the sequences generated by Algorithm 1. Under Assumptions 2 and 3, if we choose $\alpha_k = \Theta(K^{-\frac{9}{10}})$, $\beta_k = \Theta(K^{-\frac{1}{2}})$, $\gamma_k = \Theta(K^{-\frac{2}{5}})$, and $\rho = \Theta(K^{-\frac{1}{5}})$, it holds $\forall x^* \in \mathcal{X}$ that*

$$\frac{1}{K}\sum_{k=1}^{K} \mathbb{E}[\lambda^*(x_k) \cdot (F(x_k) - F(x^*))] = \mathcal{O}\left(K^{-\frac{1}{10}}\right). \tag{17}$$

If we choose $x^*$ as the Pareto-optimal point and $\lambda^*(x_k) > 0$, Theorem 1 captures the convergence to the Pareto-optimal objective values.

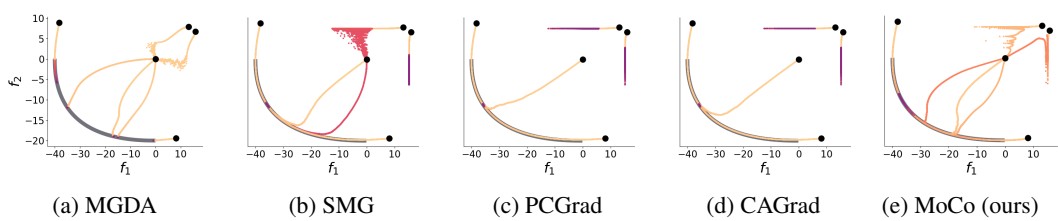

Figure 2: Comparison of trajectories in the objective space. We use five initializations in the same toy example in Figure 1, and plot the optimization trajectory in the objective space. MGDA converges to the Pareto front from all of the initializations. SMG, PCGrad, and CAGrad which only have access to single stochastic gradient per objective fail to converge to the Pareto front in some initializations. Our MoCo follows a similar trajectory to that of MGDA, and finds the Pareto front for each initialization.

In many practical problems the objective functions are non-convex, and the following theorem establishes the convergence of the proposed method for non-convex functions.

**Theorem 2.** *Consider the sequences generated by Algorithm 1 with $\mathcal{X} = \mathbb{R}^d$. Under Assumptions 1 and 3, if we choose $\alpha_k = \Theta(K^{-\frac{9}{10}})$, $\beta_k = \Theta(K^{-\frac{1}{2}})$, $\gamma_k = \Theta(K^{-\frac{2}{5}})$, and $\rho = \Theta(K^{-\frac{1}{5}})$, it holds*

$$\frac{1}{K} \sum_{k=1}^{K} \mathbb{E}\left[\|\nabla F(x_k)\lambda^*(x_k)\|^2\right] = \mathcal{O}\left(K^{-\frac{1}{10}}\right). \tag{18}$$

Theorem 2 shows that the MGDA direction $\nabla F(x_k)\lambda^*(x_k)$ converges to 0, which indicates that the proposed method is able to achieve Pareto-stationarity. It is the first finite-time convergence guarantee for the stochastic MGDA method under non-convex objective functions.

**Theorem 3.** *Consider the sequences generated by Algorithm 1 with $\mathcal{X} = \mathbb{R}^d$. Furthermore assume there exists a constant $F > 0$ such that for all $k \in [K]$, $\|F(x_k)\| \leq F$. Then, under Assumptions 1 and 3, if we choose $\alpha_k = \Theta(K^{-\frac{3}{5}})$, $\beta_k = \Theta(K^{-\frac{2}{5}})$, $\gamma_k = \Theta(K^{-1})$, and $\rho = 0$, it holds that*

$$\frac{1}{K} \sum_{k=1}^{K} \mathbb{E}\left[\|\nabla F(x_k)\lambda^*(x_k)\|^2\right] = \mathcal{O}\left(MK^{-\frac{2}{5}}\right). \tag{19}$$

Theorem 3 shows Algorithm 1 will converge to a Pareto stationary point with an improved convergence rate, if the sequence of functions $F(x_1), F(x_2), \ldots, F(x_k)$ are bounded.

**Remark 3** (Comparison with SMG). *Theorems 1 and 2 provide the convergence rates of MoCo under Assumptions 1–3 and Assumptions 1 and 3, respectively. Compared to the convergence analysis of SMG in (Liu & Vicente, 2021), the convergence rates in Theorems 1 and 2 are derived under small batch size, without the unjustified assumption on the Lipschitz continuity of $\lambda^*(x)$ and additionally account for the non-convex MOO setting. This may not be true unless $\nabla F(x)$ is full rank which can not be true at Pareto stationary points. We overcome this problem by adding a properly chosen regularization to the problem. We also provide an improved sample/iteration complexity with Theorem 3 under some additional assumptions on $F(x_k)$. Furthermore, we provide an improvement over Theorem 3 with a modified assumption on the stochastic gradient bias in Appendix J.*

## 4 RELATED WORK

To put our work in context, we review prior art that we group in the following two categories.

**Multi-task learning.** MTL algorithms find a common model that can solve multiple possibly related tasks. MTL has shown great success in many fields such as natural language processing, computer vision and robotics (Hashimoto et al., 2016), (Ruder, 2017), (Zhang & Yang, 2021), (Vandenhende et al., 2021). One line of research involves designing machine learning models that facilitate MTL, such as architectures with task specific modules (Misra et al., 2016), with attention based mechanisms (Rosenbaum et al., 2017), (Yang et al., 2020), or with different path activation corresponding to different tasks. Our method is model agnostic, and thus can be applied to these methods in a complementary manner. Another line of work focuses on decomposing a problem into multiple local

| Method | Segmentation (Higher Better) | | Depth (Lower Better) | | $\Delta m\% \downarrow$ |
|---|---|---|---|---|---|
| | mIoU | Pix Acc | Abs Err | Rel Err | |
| Independent | 74.01 | 93.16 | 0.0125 | 27.77 | - |
| Cross-Stitch (Misra et al., 2016) | 73.08 | 92.79 | 0.0165 | 118.5 | 90.02 |
| MTAN (Liu et al., 2019) | 75.18 | 93.49 | 0.0155 | 46.77 | 22.60 |
| MGDA (Sener & Koltun, 2018) | 68.84 | 91.54 | 0.0309 | **33.50** | 44.14 |
| PCGrad (Yu et al., 2020a) | 75.13 | 93.48 | 0.0154 | 42.07 | 18.29 |
| GradDrop (Chen et al., 2020) | 75.27 | 93.53 | 0.0157 | 47.54 | 23.73 |
| CAGrad (Liu et al., 2021a) | 75.16 | 93.48 | **0.0141** | 37.60 | 11.64 |
| **MoCo (ours)** | **75.42** | **93.55** | 0.0149 | 34.19 | **9.90** |

Table 1: Multi-task supervised learning on CityScape dataset with the 7-class semantic segmentation and depth estimation results. Results are averaged over 3 independent runs. CAGrad, PCGrad, GradDrop and our method are applied on the MTAN backbone.

tasks and learn these tasks using smaller models (Rusu et al., 2015),(Parisotto et al., 2015), (Teh et al., 2017), (Ghosh et al., 2017). These models are then aggregated into a single model using knowledge distillation (Hinton et al., 2015). Our method does not require multiple models in learning, and focus on learning different tasks simultaneously using a single model.

**Gradient-based MOO.** This line of work involves optimizing multiple objectives simultaneously using gradient manipulations. A foundational algorithm in this regard is MGDA(Désidéri, 2012), which dynamically combine gradients to find a common descent direction for all objectives. A comprehensive convergence analysis for the deterministic MGDA algorithm has been provided in (Fliege et al., 2019). Recently, (Liu & Vicente, 2021) extends this analysis to the stochastic counterpart of multi-gradient descent algorithm, for smooth convex and strongly convex functions. However, this work makes strong assumptions on the bias of the stochastic gradient and does not consider the nested MOO setting that is central to the multi-task reinforcement learning. In Yang et al. (2021), the authors establish convergence of stochastic MGDA under the assumption of access to true convex combination coefficients, which may not be true in a practical stochastic optimization setting. Another related line of work considers the optimization challenges related to MTL, considering task losses as objectives. One common approach is to find gradients for balancing learning of different tasks. The simplest way is to re-weight per task losses based on a specific criteria such as uncertainty (Kendall et al., 2017), gradient norms (Chen et al., 2018) or task difficulty (Guo et al., 2018). These methods are often heuristics and may be unstable. More recent work (Sener & Koltun, 2018), (Yu et al., 2020a), (Liu et al., 2021a), (Gu et al., 2021) introduce gradient aggregation methods which mitigate conflict among tasks while preserving utility. In (Sener & Koltun, 2018), MTL has been first tackled through the lens of MOO techniques using MGDA. In (Yu et al., 2020a), a new method called PCGrad has been developed to amend gradient magnitude and direction in order to avoid conflicts among per task gradients. In (Liu et al., 2021a), an algorithm similar to MGDA, named CAGrad, has been developed, which uniquely minimizes the average task loss. In (Liu et al., 2021b), an impartial objective gradient modification mechanism has been studied. Concurrent to our work, a Nash bargaining solution has been proposed in (Navon et al., 2022) for weighting per objective gradients. All the aforementioned works on MTL use the deterministic objective gradient for analysis (if any), albeit the accompanying empirical evaluations are done in a stochastic setting. There are also gradient-based MOO algorithms that find a set of Pareto optimal points for a given problem rather than one. To this end, works such as (Liu et al., 2021c; Liu & Vicente, 2021; Lin et al., 2019; Mahapatra & Rajan, 2021; Navon et al., 2020; Lin et al., 2022; Kyriakis et al., 2021; Yang et al., 2021; Zhao et al., 2021; Momma et al., 2022), develop algorithms that find multiple points in the Pareto front in mutil-task supervised learning or reinforecement learning settings, ensuring some quality of the obtained set of Pareto points. Our work is orthogonal to this line of research, and can potentially be combined with those method to achieve better performance.

## 5 EXPERIMENTS

In this section, first we provide further illustration of our method in comparison with existing gradient-based MOO algorithms in the toy example introduced in Section 1. Then we provide empirical

| Method | Segmetation | | Depth | | Surface Normal | | | | | $\Delta m\%\downarrow$ |
|---|---|---|---|---|---|---|---|---|---|---|
| | (Higher Better) | | (Lower Better) | | Angle Distance (Lower Better) | | Within $t^\circ$ (Higher better) | | | |
| | mIoU | Pix Acc | Abs Err | Rel Err | Mean | Median | 11.25 | 22.5 | 30 | |
| Independent | 38.30 | 63.76 | 0.6754 | 0.2780 | 25.01 | 19.21 | 30.14 | 57.20 | 69.15 | - |
| Cross-Stitch | 37.42 | 63.51 | 0.5487 | **0.2188** | 28.85 | 24.52 | 22.75 | 46.58 | 59.56 | 6.96 |
| MTAN | 39.29 | 65.33 | 0.5493 | 0.2263 | 28.15 | 23.96 | 22.09 | 47.50 | 61.08 | 5.59 |
| MGDA | 30.47 | 59.90 | 0.6070 | 0.2555 | **24.88** | **19.45** | **29.18** | **56.88** | **69.36** | 1.38 |
| PCGrad | 38.06 | 64.64 | 0.5550 | 0.2325 | 27.41 | 22.80 | 23.86 | 49.83 | 63.14 | 3.97 |
| GradDrop | 39.39 | 65.12 | **0.5455** | 0.2279 | 27.48 | 22.96 | 23.38 | 49.44 | 62.87 | 3.58 |
| CAGrad | 39.79 | 65.49 | 0.5486 | 0.2250 | 26.31 | 21.58 | 25.61 | 52.36 | 65.58 | 0.20 |
| **MoCo (ours)** | **40.30** | **66.07** | 0.5575 | 0.2135 | 26.67 | 21.83 | 25.61 | 51.78 | 64.85 | **0.16** |

Table 2: Multi-task supervised learning on NYU-v2 dataset with 13-class semantic segmentation, depth estimation, and surface normal prediction results on NYU-v2 dataset. Results are averaged over 3 independent runs. CAGrad, PCGrad, GradDrop and MoCo are applied on the MTAN backbone.

comparison of our proposed method with the state-of-the-art MTL algorithms, using challenging and widely used real world MTL benchmarks in supervised and reinforcement learning settings. The details of hyperparameters are provided in Appendix K.

**Toy example.** To further elaborate on how MoCo converges to a Pareto stationary point, we again optimize the two objectives given in Figure 1 and demonstrate the performance in the objective space (Figure 2). MGDA with true gradients converges to a Pareto stationary point in all initializations. However, it can be seen that SMG, PCGrad,and CAGrad methods fail to converge to a Pareto stationary point, and end up in dominated points in the objective space for some initializations. This is because these algorithms use a biased multi-gradient that does not become zero. In contrast, MoCo converges to Pareto stationary points in every initialization, and follows a similar trajectory to MGDA.

## 5.1 SUPERVISED LEARNING

We compare MoCo with existing MTL algorithms using NYU-v2 (Silberman et al., 2012) and CityScapes (Cordts et al., 2015) datasets. We follow the experiment setup of (Liu et al., 2021a) and combine our method with MTL method MTAN (Liu et al., 2019), which applies an attention mechanism. We evaluate our method in comparison to CAGrad, PCGrad, vanilla MTAN and Cross-Stitch (Misra et al., 2016). Following (Maninis et al., 2019; Liu et al., 2021a; Navon et al., 2022), we use the per-task performance drop of a metric $S_m$ for method $\mathcal{A}$ with respect to baseline $\mathcal{B}$ as a measure of the overall performance of a given method, which is given by $\Delta m = \frac{1}{M}\sum_{m=1}^{M}(-1)^{\ell_m}(S_{\mathcal{A},m} - S_{\mathcal{B},m})/S_{\mathcal{B},m}$ , where $M$ is the number of tasks, $S_{\mathcal{B},m}$ and $S_{\mathcal{A},m}$ are the values of metric $S_m$ obtained by the baseline and the compared method respectively. Here, $\ell_m = 1$ if higher values for $S_m$ are better and 0 otherwise.

The results of the experiments are shown in Table 1 and 2. Our method, MoCo, outperforms all the existing MTL algorithms in terms of $\Delta m\%$ for both Cityscapes and NYU-v2 datasets. Since our method focuses on gradient correction, our method can also be applied on top of existing gradient-based MOO methods. Additional experiments regarding this are provided in Appendix K.

## 5.2 REINFORCEMENT LEARNING

For the multi-task reinforcement learning setting, we use the multi-task reinforcement learning benchmark MT10 available in the Met-world environment (Yu et al., 2020b). We follow the experimental setup used in (Liu et al., 2021a) and provide the empirical comparison between our MoCo method and the existing baselines. Specifically, we use MTRL codebase (Sodhani & Zhang, 2021) and use soft actor-critic (SAC) (Haarnoja et al., 2018) as the underlying reinforcement learning algorithm. Due to space limitation, the experiment results in a multi-task reinforcement learning setting and details of hyperparameter selection are provided in Appendix K.

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

## A    NOTATIONS

In this section we summarize the notations used in the paper and a corresponding brief description.

| Notation | Description |
|---|---|
| $x$ | Decision variable / model parameter |
| $\xi$ | Some random variable independent of $x$ |
| $\mathcal{X}$ | Feasible set of $x$ |
| $M$ | Number of objectives in the MOO problem |
| $f_m(x)$ | An objective such that $f_m : \mathcal{X} \to \mathbb{R}$, where $m \in \{1, 2, \ldots, M\}$ |
| $F(x)$ | Vector of functions containing $f_1(x), f_2(x), \ldots, f_M(x)$, such that $F : \mathcal{X} \to \mathbb{R}^M$ |
| $\nabla F(x)$ | Jacobian of $F(x)$, wich has $\nabla f_1(x), \nabla f_2(x), \ldots, \nabla f_M(x)$ as columns |
| $R_{++}^M$ | Positive othant cone of dimension $M$ |
| $\Delta^M$ | $M$ dimensional probability simplex |
| $\mathbf{1}$ | $M$ dimensional vector of ones |
| $\lambda$ | An element of $\Delta^M$ |
| $\lambda^*(x)$ | A solution to the problem (2) (original MGDA subproblem) |
| $\lambda^g(x)$ | A solution to the problem (4) (SMG subproblem) |
| $\lambda_\rho^*(x)$ | The unique solution to the problem (8) ($\ell_2$ regularized sub-problem) |
| $d(x)$ | MGDA mult-gradient given by (36) |
| $x_k$ | Model parameter at iteration $k$, updated as (10) |
| $h_{k,m}$ | Stochastic gradient estimator of $\nabla f_m(x_k)$ |
| $y_{k,m}$ | "Tracking" variable that approximates $\nabla f_m(x_k)$ |
| $Y_k$ | Matrix containing $y_{k,1}, y_{k,2}, \ldots, y_{k,M}$ as columns, updated as (6) |
| $\lambda_k$ | Estimate for $\lambda_\rho^*(x_k)$, updated as (9) |
| $\alpha_k, \beta_k, \gamma_k$ | Learning rates of $x_k, y_{k,m}$, and $\lambda_k$ updates, respectively |
| $\rho$ | $\ell_2$ regularization parameter used in (8) |
| Extension to bi-level optimization setting | |
| $z$ | Lower level parameter where $z \in \mathbb{R}^d$ |
| $l_m$ | Lower level objective such that $f_m : \mathcal{X} \to \mathbb{R}$, where $m \in \{1, 2, \ldots, M\}$ |
| $\varphi$ | Some random variable independent of $x, z$ |
| $z_m^*(x)$ | Minimizer of $l_m(x, z)$ with respect to $z$ for any $x \in \mathcal{X}$ |
| $z_{k,m}$ | Approximation of $z_m^*(x_k)$, updated as equation 13 |
| $\nabla_{zz}^2 l_m(x, z)$ | Hessian of $l_m(x, z)$ w.r.t. to $z$ |
| $H_{k,m}^{zz}$ | Stochastic approximation of $\nabla_{zz}^2 l_m(x_k, z_{k,m})$ |

Table 3: Some important notations used in the paper

## B    ADDITIONAL RELATED WORK

In this section we provide brief discussion on additional work related to MTL and MOO. We first discuss a closely related concurrent work Zhou et al. (2022), which was not available at the time of submission of this paper. In Zhou et al. (2022), the authors introduce a unified framework for stochastic gradient-based MOO algorithms. Similar to MoCo, the proposed algorithm in Zhou et al. (2022) also uses momentum-like techniques to reduce the multi-gradient bias. The key algorithmic difference of the aforementioned method compared to MoCo is that MoCo uses momentum-based correction on the full gradient estimation while Zhou et al. (2022) uses the momentum-like moving averaging for convex combination coefficients $\lambda$ computed using stochastic gradients. As a consequence of this difference, it is not clear if the averaged weight $\lambda$ obtained in Zhou et al. (2022) will ensure convergence of the stochastic MOO update direction to any deterministic MOO direction. On the other hand, in our work, we can show the convergence to MGDA direction in Lemma 2. In terms of theoretical results in Zhou et al. (2022), the authors provide convergence guarantees of the proposed algorithm for both convex and non-convex cases with convergence rates similar to that of single objective stochastic gradient descent (SGD). However, in order to obtain these results, Zhou et al. (2022) requires additional assumptions on bound of the function values, which is not required in standard single objective SGD convergence analysis. On the other hand, in this work, the convergence

guarantees presented in Theorems 1 and 2 are based on fairly standard assumptions in optimization literature, and does not assume any bound on the function value. Furthermore, the results are based on weaker assumptions on the stochastic gradient bias which facilitates bi-level MOO. In addition, we provide improved convergence results for MoCo in Theorems 3 and 4, with stronger assumptions on function value bounds similar to those in Zhou et al. (2022), but still with weaker assumptions on the bias of stochastic gradients.

In addition to gradient based MOO which is the main focus of this paper, there also exist non gradient based blackbox MOO algorithms such as (Deb et al., 2002; Golovin & Zhang, 2020; Knowles, 2006; Konakovic Lukovic et al., 2020), which are based on an evolutionary algorithm or Bayesian optimization. However, these methods often suffer from the curse of dimensionality, and may not be feasible in large scale MOO problems.

On the other hand, recent works in MTL have analyzed MTL from different viewpoints. In (Wang et al., 2021), the authors explore the connection between gradient based meta learning and MTL. In (Ye et al., 2021), the meta learning problem with multiple objectives in the upper levels has been tackled via a gradient based MOO approach. The importance of task grouping in MTL is analysed in works such as (Fifty et al., 2021). In (Meyerson & Miikkulainen, 2020), the authors show that seemingly unrelated tasks can be used for MTL.

## C   SUMMARY OF COMPARISON WITH CLOSELY RELATED PRIOR WORK

| Method | Batch size | Non-convex | Lipschitz continuity of $\lambda^*(x)$ | Bounded functions | Biased gradient | Sample complexity |
|---|---|---|---|---|---|---|
| SMG (Theorem 1) | $\mathcal{O}\left(\epsilon^{-2}\right)$ | ✗ | ✓ | ✗ | ✗ | $\mathcal{O}\left(\epsilon^{-4}\right)$ |
| MoCo (Theorem 1) | $\mathcal{O}(1)$ | ✗ | ✗ | ✗ | $\mathcal{O}\left(\frac{\alpha^2}{\beta^2}\right)$ | $\mathcal{O}\left(\epsilon^{-10}\right)$ |
| MoCo (Theorem 2) | $\mathcal{O}(1)$ | ✓ | ✗ | ✗ | $\mathcal{O}\left(\frac{\alpha^2}{\beta^2}\right)$ | $\mathcal{O}\left(\epsilon^{-10}\right)$ |
| MoCo (Theorem 3) | $\mathcal{O}(1)$ | ✓ | ✗ | ✓ | $\mathcal{O}\left(\frac{\alpha^2}{\beta^2}\right)$ | $\mathcal{O}\left(\epsilon^{-2.5}\right)$ |
| MoCo (Theorem 4) | $\mathcal{O}(1)$ | ✓ | ✗ | ✓ | $\mathcal{O}\left(\beta\right)$ | $\mathcal{O}\left(\epsilon^{-2}\right)$ |

Table 4: Comparison of MoCo with prior work on gradient based stochastic MOO, stochastic multi-gradient method (SMG)(Liu & Vicente, 2021). Here the "Batch size" column represents the number of samples used at each (outer level) iteration, "Non-convex" column denotes whether the analysis is valid for non-convex functions, "Lipschitz continuity of $\lambda^*(x)$" column denotes whether Lipschitz continuity of $\lambda^*(x)$ (see Remark 1) with respect to $x$ was assumed, "Bounded functions" column denotes whether boundedness of function values was assumed, "Biased gradients" column denotes bias in the stochastic gradient of ofunctions allowed in analysis, and "Sample complexity" column provides the (outer level) sample complexity of the corresponding method.

## D   ALGORITHM FOR MOCO WITH INEXACT GRADIENT

In this section we provide the omitted pseudo-code for MoCo with inexact gradients described in Section 3.2. As remarked in Section 3.2, this algorithm can relate to the actor critic setting with multiple critics. In Appendix K.3, we provide empirical evaluations of multi-task reinforcement learning using soft actor critic.

## E   PROOF OF LEMMA 1

Throughout the section, we write $\mathbb{E}[\cdot|\mathcal{F}_k]$ as $\mathbb{E}_k[\cdot]$ for conciseness. Consider the following conditions

(a) For any $x \in \mathcal{X}$, $l_m(x,z)$ is strongly convex w.r.t. $z$ with modulus $\mu_m > 0$.

---

**Algorithm 2** MoCo with inexact gradient

---

1: **Input** Initial model parameter $x_0$, tracking parameters $\{y_{0,i}\}_{m=1}^M$, lower level parameter $\{z_{0,i}\}_{m=1}^M$, convex combination coefficient parameter $\lambda_0$, and their respective learning rates $\{\alpha_k\}_{k=0}^K$, and $\{\beta_k\}_{k=0}^K$, $\{\gamma_k\}_{k=0}^K$.
2: **for** $k = 0, \ldots, K-1$ **do**
3:     **for** objective $m = 1, \ldots, M$ **do**
4:         Update $z_{k+1,m}$ following (13)
5:         Obtain $h_{k,m}$ following (14)
6:         Update $y_{k+1,m}$ following (6)
7:     **end for**
8:     Update $\lambda_{k+1}$ and $x_{k+1}$ following (9)-(10)
9: **end for**
10: **Output** $x_K$

---

(b) There exist constants $L_{xz}, l_{xz}, l_{zz}$ such that $\nabla_z l_m(x, z)$ is $L_{xz}$-Lipschitz continuous w.r.t. $x$; $\nabla_z l_m(x, z)$ is $L_{zz}$-Lipschitz continuous w.r.t. $z$. $\nabla_{xz} l_m(x, z), \nabla_{zz} l_m(x, z)$ are respectively $l_{xz}$-Lipschitz and $l_{zz}$-Lipschitz continuous w.r.t. $(x, z)$.

(c) There exist constants $l_{fx}, l_{fz}, l'_{fz}, l_z$ such that $\nabla_x f_m(x, z)$ and $\nabla_z f_m(x, z)$ are respectively $l_{fx}$ and $l_{fz}$ Lipschitz continuous w.r.t. $z$; $\nabla_z f_m(x, z)$ is $l'_{fz}$-Lipschitz continuous w.r.t. $x$; $f_m(x, z)$ is $l_z$-Lipschitz continuous w.r.t. $z$.

(d) There exist constants $C_H$ and $\sigma_H$ such that $\|\mathbb{E}_k[H_{k,m}^{zz}] - [\nabla_{zz} l_m(x_k, z_{k,m})]^{-1}\|^2 \leq C_H \alpha_k^2/\beta_k^2$ and $\mathbb{E}_k\|H_{k,m}^{zz}\|^2 \leq \sigma_H^2$. There exist constants $C_x, C_{xz}, C_z, C_l$ such that $\mathbb{E}_k\|\nabla_x f_m(x_k, z_{k,m}, \xi_k)\|^2 \leq C_x^2$, $\mathbb{E}_k\|\nabla_{xz} l_m(x_k, z_{k,m}, \varphi'_k)\|^2 \leq C_{xz}^2$, $\mathbb{E}_k\|\nabla_z f_m(x_k, z_{k,m}, \xi_k)\|^2 \leq C_z^2$ and $\mathbb{E}_k\|\nabla_z l_m(x_k, z_{k,m}, \varphi_k)\|^2 \leq C_l^2$.

The above conditions are standard in the literature (Ghadimi & Wang, 2018). In particular, condition (d) on $H_{k,m}^{zz}$ can be guaranteed by (Ghadimi & Wang, 2018, Algorithm 3). With these conditions, we first give a restatement of Lemma 1.

**Lemma 3** (Restatement of Lemma 1). *Consider the sequences generated by (13), (6), (9) and (10). Under conditions (a)–(d), if we choose the step sizes as those of Lemma 2, we have for any $m$ that*

$$\frac{1}{K} \sum_{k=1}^K \mathbb{E}\|\mathbb{E}[h_{k,m}|\mathcal{F}_k] - \nabla f_m(x_k)\|^2 = \mathcal{O}\left(\frac{\alpha_K^2}{\beta_K^2}\right). \tag{20}$$

*There exists a constant $\sigma_0$ such that*

$$\mathbb{E}[\|h_{k,m} - \mathbb{E}[h_{k,m}|\mathcal{F}_k]\|^2|\mathcal{F}_k] \leq \sigma_0^2. \tag{21}$$

*Proof.* We first prove (21) with

$\mathbb{E}_k\|h_{k,m} - \mathbb{E}_k[h_{k,m}]\|^2$

$= \mathbb{E}_k\|h_{k,m}\|^2 - \|\mathbb{E}_k[h_{k,m}]\|^2 \leq \mathbb{E}_k\|h_{k,m}\|^2$

$\leq 2\mathbb{E}_k\|\nabla_x f_m(x_k, z_{k,m}, \xi_k)\|^2 + 2\mathbb{E}_k\left[\|\nabla_{xz}^2 l_m(x_k, z_{k,m}, \varphi'_k)\|^2 \|H_{k,m}^{zz}\|^2 \|\nabla_z f_m(x_k, z_{k,m}, \xi_k)\|^2\right]$

$\leq 2C_x^2 + 2C_{xz}^2 C_z^2 \sigma_H^2, \tag{22}$

where the last inequality follows from item (d) along with the independence of $\nabla_{xz}^2 l_m(x_k, z_{k,m}, \varphi'_k)$, $H_{k,m}^{zz}$ and $\nabla_z f_m(x_k, z_{k,m}, \xi_k)$ given $\mathcal{F}_k$.

Next we start to prove (20).

$$\mathbb{E}_k[h_{k,m}] = \nabla_x f_m(x_k, z_{k,m}) - \nabla_{xz}^2 l_m(x_k, z_{k,m}) \mathbb{E}_k[H_{k,m}^{zz}] \nabla_z f_m(x_k, z_{k,m}). \tag{23}$$

In the following proof, we write $z_{k,m}^*(x_k)$ as $z_{k,m}^*$. The above inequality along with (12) implies

$$\|\mathbb{E}_k[h_{k,m}] - \nabla f_m(x_k)\|$$
$$\leq \|\nabla_x f_m(x_k, z_{k,m}) - \nabla_x f_m(x_k, z_{k,m}^*)\|$$
$$+ \|\nabla_{xz}^2 l_m(x_k, z_{k,m})\mathbb{E}_k[H_{k,m}^{zz}]\nabla_z f_m(x_k, z_{k,m}) - \nabla_{xz}^2 l_m(x_k, z_{k,m})[\nabla_{zz} l_m(x_k, z_{k,m})]^{-1}\nabla_z f_m(x_k, z_{k,m})\|$$
$$+ \|\nabla_{xz}^2 l_m(x_k, z_{k,m})[\nabla_{zz} l_m(x_k, z_{k,m})]^{-1}\nabla_z f_m(x_k, z_{k,m})$$
$$- \nabla_{xz}^2 l_m(x_k, z_{k,m}^*)[\nabla_{zz} l_m(x_k, z_{k,m}^*)]^{-1}\nabla_z f_m(x_k, z_{k,m}^*)\|$$
$$\leq l_{fx}\|z_{k,m} - z_m^*(x_k)\| + L_{xz}l_z\|\mathbb{E}_k[H_{k,m}^{zz}] - [\nabla_{zz} l_m(x_k, z_{k,m})]^{-1}\|$$
$$+ \big(\frac{L_{xz}l_z}{\mu_m} + \frac{l_{fz}}{\mu_m} + \frac{l_{fz}l_zl_{zz}}{\mu_m^2}\big)\|z_{k,m} - z_m^*(x_k)\| \tag{24}$$

where the last inequality follows from conditions (a)–(c). The inequality (24) implies

$$\|\mathbb{E}_k[h_{k,m}] - \nabla f_m(x_k)\|^2 \leq 2L_{xz}^2 l_z^2\|\mathbb{E}_k[H_{k,m}^{zz}] - [\nabla_{zz} l_m(x_k, z_{k,m})]^{-1}\|^2$$
$$+ 2\big(l_{fx} + \frac{L_{xz}l_z}{\mu_m} + \frac{l_{fz}}{\mu_m} + \frac{l_{fz}l_zl_{zz}}{\mu_m^2}\big)^2\|z_{k,m} - z_m^*(x_k)\|^2$$
$$\leq 2L_{xz}^2 l_z^2 C_H \frac{\alpha_K^2}{\beta_K^2} + 2\big(l_{fx} + \frac{L_{xz}l_z}{\mu_m} + \frac{l_{fz}}{\mu_m} + \frac{l_{fz}l_zl_{zz}}{\mu_m^2}\big)^2\|z_{k,m} - z_m^*(x_k)\|^2 \tag{25}$$

where the last inequality follows from condition (d) on the quality of $H_{k,m}^{zz}$. Thus to prove (20), it suffices to prove $\frac{1}{K}\sum_{k=1}^K \mathbb{E}\|z_{k,m} - z_{k,m}^*\|^2 = \mathcal{O}\big(\frac{\alpha_K^2}{\beta_K^2}\big)$ next.

*The convergence of $z_{k,m}$.* We start with

$$\|z_{k+1,m} - z_{k+1,m}^*\|^2 = \|z_{k+1,m} - z_{k,m}^*\|^2 + 2\langle z_{k+1,m} - z_{k,m}^*, z_{k,m}^* - z_{k+1,m}^*\rangle + \|z_{k,m}^* - z_{k+1,m}^*\|^2. \tag{26}$$

The first term is bounded as

$$\mathbb{E}_k\|z_{k+1,m} - z_{k,m}^*\|^2$$
$$= \mathbb{E}_k\|z_{k,m} - \beta_k\nabla_z l_m(x_k, z_{k,m}, \varphi_k) - z_{k,m}^*\|^2$$
$$= \|z_{k,m} - z_{k,m}^*\|^2 - 2\beta_k\langle z_{k,m} - z_{k,m}^*, \nabla_z l_m(x_k, z_{k,m})\rangle + \beta_k^2\mathbb{E}_k\|\nabla_z l_m(x_k, z_{k,m}, \varphi_k)\|^2$$
$$\leq (1 - 2\mu_m\beta_k)\|z_{k,m} - z_{k,m}^*\|^2 + C_l^2\beta_k^2 \tag{27}$$

where the last inequality follows from condition (a) and (d).

Under conditions (a)–(c), it is shown that there exists a constant $L_{z,m}$ such that $z_m^*(x)$ is $L_{z,m}$-lipschitz continuous (Ghadimi & Wang, 2018)[Lemma 2.2 (b)]. Let $L_z = \max_m L_{z,m}$, then the second term in (26) can be bounded as

$$\langle z_{k+1,m} - z_{k,m}^*, z_{k,m}^* - z_{k+1,m}^*\rangle \leq L_z\|z_{k+1,m} - z_{k,m}^*\|\|x_k - x_{k+1}\|$$
$$\leq L_z C_y\alpha_k\|z_{k+1,m} - z_{k,m}^*\|$$
$$\leq \frac{\mu_m}{2}\beta_k\|z_{k+1,m} - z_{k,m}^*\|^2 + \frac{1}{2}L_z^2 C_y^2\mu_m^{-1}\frac{\alpha_k^2}{\beta_k} \tag{28}$$

where $C_y = \sup\|Y_k\| = \sup_{x\in\mathcal{X}}\|\nabla F(x)\| < \infty$ (by Assumption 1 or 2 and projection), and the last inequality uses the Young's inequality.

The last term in (26) is bounded as

$$\|z_{k,m}^* - z_{k+1,m}^*\|^2 \leq L_z^2 C_y^2\alpha_k^2. \tag{29}$$

Substituting (27)–(29) into (26) yields

$$\mathbb{E}_k\|z_{k+1,m} - z_{k+1,m}^*\|^2 \leq (1 - \mu_m\beta_k)\|z_{k,m} - z_{k,m}^*\|^2 + C_l^2\beta_k^2 + L_z^2 C_y^2\mu_m^{-1}\frac{\alpha_k^2}{\beta_k} + L_z^2 C_y^2\alpha_k^2. \tag{30}$$

Taking total expectation on both sides, and then telescoping implies that (set $\alpha_k = \alpha_K, \beta_k = \beta_K, \forall k$)

$$\frac{1}{K} \sum_{k=1}^{K} \mathbb{E}\|z_{k,m} - z_{k,m}^*\|^2 = \mathcal{O}\left(\frac{1}{K\beta_K}\right) + \mathcal{O}(\beta_K) + \mathcal{O}(\alpha_K) + \mathcal{O}\left(\frac{\alpha_K^2}{\beta_K^2}\right). \tag{31}$$

The last inequality along with the step size choice in Lemma 3 implies that

$$\frac{1}{K} \sum_{k=1}^{K} \mathbb{E}\|z_{k,m} - z_{k,m}^*\|^2 = \mathcal{O}\left(\frac{\alpha_K^2}{\beta_K^2}\right), \tag{32}$$

which along with (25) implies (20). This completes the proof. $\qquad\square$

## F  PROOF OF LEMMA 2

Before we present the main proof, we first introduce the Lemmas 4 and 5, which are direct consequences of (Dontchev & Rockafellar, 2009, Theorem 2F.7) and (Koshal et al., 2011, Lemma A.1), respectively.

**Lemma 4.** *Under Assumption 1 , there exists a constant $L_\lambda := \rho^{-1} \sum_{m=1}^{M} L_m$ such that the following inequality holds*

$$\|\lambda_\rho^*(x) - \lambda_\rho^*(x')\| \le L_\lambda \|\nabla F(x) - \nabla F(x')\|, \tag{33}$$

*which further indicates*

$$\|\lambda_\rho^*(x) - \lambda_\rho^*(x')\| \le L_{\lambda,F} \|x - x'\|, \tag{34}$$

*where $L_{\lambda,F} := \rho^{-1} L$, with $L = \left(\sum\limits_{m=1}^{M} L_m\right) \left(\sum\limits_{m=1}^{M} L_{m,1}\right)$.*

**Lemma 5.** *For any $\rho > 0$ and $x \in \mathcal{X}$, we have*

$$0 \le \|\nabla F(x)\lambda_\rho^*(x)\|^2 - \|\nabla F(x)\lambda^*(x)\|^2 \le \frac{\rho}{2}\left(1 - \frac{1}{M}\right). \tag{35}$$

Now we start to prove Lemma 2.

*Proof.* Throughout the following proof, we write $\mathbb{E}[\cdot|\mathcal{F}_k]$ as $\mathbb{E}_k[\cdot]$ for conciseness. With $d(x_k) = \nabla F(x_k)\lambda^*(x_k)$, we have

$$\|d(x_k) - Y_k\lambda_k\|^2$$
$$= \|\nabla F(x_k)\lambda^*(x_k) - \nabla F(x_k)\lambda_\rho^*(x_k) + \nabla F(x_k)\lambda_\rho^*(x_k) - Y_k\lambda_\rho^*(x_k) + Y_k\lambda_\rho^*(x_k) - Y_k\lambda_k\|^2$$
$$\le 3\|\nabla F(x_k)(x_k)\lambda^*(x_k) - \nabla F(x_k)\lambda_\rho^*(x_k)\|^2 + 3\|\nabla F(x_k)\lambda_\rho^*(x_k) - Y_k\lambda_\rho^*(x_k)\|^2 + 3\|Y_k\lambda_\rho^*(x_k) - Y_k\lambda_k\|^2$$
$$\le 3\|\nabla F(x_k)\lambda^*(x_k) - \nabla F(x_k)\lambda_\rho^*(x_k)\|^2 + 3\|\nabla F(x_k) - Y_k\|^2 + 3C_y\|\lambda_\rho^*(x_k) - \lambda_k\|^2. \tag{36}$$

From (36), to prove Lemma 2, it suffices to show $\|\nabla F(x_k)\lambda^*(x_k) - \nabla F(x_k)\lambda_\rho^*(x_k)\|$ diminishes, and establish the convergence of $Y_k$ and $\lambda_k$.

**Bounding** $\|\nabla F(x_k)\lambda^*(x_k) - \nabla F\lambda_\rho^*(x_k)\|$**.** Denoting $\lambda^*(x_k)$ as $\lambda_k^*$ and $\lambda_\rho^*(x_k)$ as $\lambda_{\rho,k}^*$, we first consider the following bound

$$\|\nabla F(x_k)\lambda_k^* - \nabla F(x_k)\lambda_{\rho,k}^*\|^2 = \|\nabla F(x_k)\lambda_k^*\|^2 + \|\nabla F(x_k)\lambda_{\rho,k}^*\|^2 - 2\langle \nabla F(x_k)\lambda_k^*, \nabla F(x_k)\lambda_{\rho,k}^* \rangle$$
$$\le \|\nabla F(x_k)\lambda_{\rho,k}^*\|^2 - \|\nabla F(x_k)\lambda_k^*\|^2$$
$$\le \frac{\rho}{2}, \tag{37}$$

where the first inequality is due to the optimality condition

$$\langle \lambda, \nabla F(x_k)^\top \nabla F(x_k)\lambda_k^* \rangle \ge \langle \lambda_k^*, \nabla F(x_k)^\top \nabla F(x_k)\lambda_k^* \rangle = \|\nabla F(x_k)\lambda_k^*\|^2$$

for any $\lambda \in \Delta^M$, and the last inequality is due to Lemma 5. With the choice of $\rho = \Theta(K^{-\frac{1}{5}})$ as required by Theorems 1 and 2, we have

$$\|\nabla F(x_k)\lambda_k^* - \nabla F(x_k)\lambda_{\rho,k}^*\|^2 = \mathcal{O}(K^{-\frac{1}{5}}). \tag{38}$$

**Convergence of $Y_k$.** With $\mathbb{E}_k[h_{k,m}] = \mathbb{E}[h_{k,m}|\mathcal{F}_k]$, we start by

$$\mathbb{E}_k\|y_{k+1,m} - \nabla f_m(x_{k+1})\|^2 = \mathbb{E}_k\|y_{k+1,m} - \nabla f_m(x_k)\|^2 + 2\mathbb{E}_k\langle y_{k+1,m} - \nabla f_m(x_k), \nabla f_m(x_k) - \nabla f_m(x_{k+1})\rangle$$
$$+ \mathbb{E}_k\|\nabla f_m(x_k) - \nabla f_m(x_{k+1})\|^2. \tag{39}$$

We bound the first term in (39) as

$$\mathbb{E}_k\|y_{k+1,m} - \nabla f_m(x_k)\|^2$$
$$\leq \mathbb{E}_k\|y_{k,m} - \beta_k(y_{k,m} - h_{k,m}) - \nabla f_m(x_k)\|^2$$
$$= \|y_{k,m} - \nabla f_m(x_k)\|^2 - 2\beta_k\langle y_{k,m} - \nabla f_m(x_k), y_{k,m} - \mathbb{E}_k[h_{k,m}]\rangle + \beta_k^2\mathbb{E}_k\|y_{k,m} - h_{k,m}\|^2$$
$$= \|y_{k,m} - \nabla f_m(x_k)\|^2 - 2\beta_k\langle y_{k,m} - \nabla f_m(x_k), y_{k,m} - \nabla f_m(x_k)\rangle$$
$$\quad - 2\beta_k\langle y_{k,m} - \nabla f_m(x_k), \nabla f_m(x_k) - \mathbb{E}_k[h_{k,m}]\rangle + \beta_k^2\mathbb{E}_k\|y_{k,m} - h_{k,m}\|^2$$
$$\leq (1 - 2\beta_k)\|y_{k,m} - \nabla f_m(x_k)\|^2 + 2\beta_k\|y_{k,m} - \nabla f_m(x_k)\|\|\nabla f_m(x_k) - \mathbb{E}_k[h_{k,m}]\| + \beta_k^2\mathbb{E}_k\|y_{k,m} - h_{k,m}\|^2$$
$$\leq (1 - \beta_k)\|y_{k,m} - \nabla f_m(x_k)\|^2 + \beta_k\|\nabla f_m(x_k) - \mathbb{E}_k[h_{k,m}]\|^2 + \beta_k^2\mathbb{E}_k\|y_{k,m} - h_{k,m}\|^2 \tag{40}$$

where the last inequality follows from young's inequality. Now, consider the last term of (40). Selecting $\beta_k$ such that $3\beta_k^2 \leq \beta_k/2$, and Assumption 3, we have

$$\beta_k^2\mathbb{E}_k\|y_{k,m} - h_{k,m}\|^2$$
$$= \beta_k^2\mathbb{E}_k\|y_{k,m} - \nabla f_m(x_k) + \nabla f_m(x_k) - \mathbb{E}_k[h_{k,m}] + \mathbb{E}_k[h_{k,m}] - h_{k,m}\|^2$$
$$\leq 3\beta_k^2\|y_{k,m} - \nabla f_m(x_k)\|^2 + 3\beta_k^2\|\nabla f_m(x_k) - \mathbb{E}_k[h_{k,m}]\|^2 + 3\beta_k^2\mathbb{E}_k\|\mathbb{E}_k[h_{k,m}] - h_{k,m}\|^2$$
$$\leq \frac{\beta_k}{2}\|y_{k,m} - \nabla f_m(x_k)\|^2 + 3\beta_k^2\|\nabla f_m(x_k) - \mathbb{E}_k[h_{k,m}]\|^2 + 3\sigma_m^2\beta_k^2. \tag{41}$$

Then, plugging in (41) in (40), we obtain

$$\mathbb{E}_k\|y_{k+1,m} - \nabla f_m(x_k)\|^2 \leq (1 - \frac{1}{2}\beta_k)\|y_{k,m} - \nabla f_m(x_k)\|^2 + \beta_k\|\nabla f_m(x_k) - \mathbb{E}_k[h_{k,m}]\|^2$$
$$+ 3\beta_k^2\|\nabla f_m(x_k) - \mathbb{E}_k[h_{k,m}]\|^2 + 3\sigma_m^2\beta_k^2. \tag{42}$$

The second term in (39) can be bounded as

$$\langle y_{k+1,m} - \nabla f_m(x_k), \nabla f_m(x_k) - \nabla f_m(x_{k+1})\rangle$$
$$\leq \|y_{k+1,m} - \nabla f_m(x_k)\|\|\nabla f_m(x_k) - \nabla f_m(x_{k+1})\|$$
$$\leq L_{m,1}\|y_{k+1,m} - \nabla f_m(x_k)\|\|x_{k+1} - x_k\|$$
$$\leq L_{m,1}C_y\alpha_k\|y_{k+1,m} - \nabla f_m(x_k)\|$$
$$\leq \frac{1}{8}\beta_k\|y_{k+1,m} - \nabla f_m(x_k)\|^2 + 2L_{m,1}^2C_y^2\frac{\alpha_k^2}{\beta_k}$$
$$\leq \frac{1}{8}\beta_k\|y_{k+1,m} - \nabla f_m(x_k)\|^2 + 2\bar{L}_1^2C_y^2\frac{\alpha_k^2}{\beta_k} \tag{43}$$

where the second last inequality follows from young's inequality, and the last inequality follows from the definition $\bar{L}_1 = \max_m L_{m,1}$.

The last term in (39) can be bounded as

$$\|\nabla f_m(x_k) - \nabla f_m(x_{k+1})\|^2 \leq \bar{L}_1^2\|x_{k+1} - x_k\|^2 \leq \bar{L}_1^2C_y^2\alpha_k^2. \tag{44}$$

Collecting the upper bounds in (42)–(44) and substituting them into (39) gives

$$\mathbb{E}_k\|y_{k+1,m} - \nabla f_m(x_{k+1})\|^2 \leq (1 - \frac{1}{4}\beta_k)\|y_{k,m} - \nabla f_m(x_k)\|^2 + (\beta_k + 3\beta_k^2)\|\nabla f_m(x_k) - \mathbb{E}_k[h_{k,m}]\|^2$$
$$+ 3\sigma_m^2\beta_k^2 + \bar{L}_1^2C_y^2\alpha_k^2 + 4\bar{L}_1^2C_y^2\frac{\alpha_k^2}{\beta_k}. \tag{45}$$

Suppose $\alpha_k$ and $\beta_k$ are constants. Taking total expectation and then telescoping both sides of (45) gives

$$\frac{1}{K}\sum_{k=1}^{K}\mathbb{E}\|y_{k,m} - \nabla f_m(x_k)\|^2 = \mathcal{O}\big(\frac{1}{K\beta_k}\big) + \mathcal{O}\big(\frac{1}{K}\sum_{k=1}^{K}\mathbb{E}\|\nabla f_m(x_k) - \mathbb{E}_k[h_{k,m}]\|^2\big)$$
$$+ \mathcal{O}(\beta_k) + \mathcal{O}\big(\frac{\alpha_k^2}{\beta_k^2}\big) \tag{46}$$

Along with the choice of step sizes as required by Theorems 1 and 2, and due to Assumption 3, the last inequality gives

$$\frac{1}{K}\sum_{k=1}^{K}\mathbb{E}\|y_{k,m} - \nabla f_m(x_k)\|^2 = \mathcal{O}\big(K^{-\frac{1}{2}}\big) \tag{47}$$

which, based on the definitions of $Y_k$ and $\nabla F(x_k)$, implies that

$$\frac{1}{K}\sum_{k=1}^{K}\mathbb{E}\|Y_k - \nabla F(x_k)\|^2 = \mathcal{O}\big(MK^{-\frac{1}{2}}\big). \tag{48}$$

**Convergence of $\lambda_k$.** We write $\lambda_\rho^*(x_k)$ in short as $\lambda_{\rho,k}^*$ in the following proof. We start by

$$\|\lambda_{k+1} - \lambda_{\rho,k+1}^*\|^2 = \|\lambda_{k+1} - \lambda_{\rho,k}^*\|^2 + 2\langle\lambda_{k+1} - \lambda_{\rho,k}^*, \lambda_{\rho,k}^* - \lambda_{\rho,k+1}^*\rangle + \|\lambda_{\rho,k}^* - \lambda_{\rho,k+1}^*\|^2. \tag{49}$$

The first term is bounded as

$$\|\lambda_{k+1} - \lambda_{\rho,k}^*\|^2 = \|\Pi_{\Delta^M}\big(\lambda_k - \gamma_k\big(Y_k^\top Y_k + \rho I\big)\lambda_k\big) - \lambda_{\rho,k}^*\|^2$$
$$\leq \|\lambda_k - \gamma_k\big(Y_k^\top Y_k + \rho I\big)\lambda_k - \lambda_{\rho,k}^*\|^2$$
$$= \|\lambda_k - \lambda_{\rho,k}^*\|^2 - 2\gamma_k\langle\lambda_k - \lambda_{\rho,k}^*, \big(Y_k^\top Y_k + \rho I\big)\lambda_k\rangle + \gamma_k^2\|\big(Y_k^\top Y_k + \rho I\big)\lambda_k\|^2$$
$$\leq \|\lambda_k - \lambda_{\rho,k}^*\|^2 - 2\gamma_k\langle\lambda_k - \lambda_{\rho,k}^*, \big(Y_k^\top Y_k + \rho I\big)\lambda_k\rangle + (C_y^2 + \rho)^2\gamma_k^2 \tag{50}$$

Consider the second term in the last inequality:

$$\langle\lambda_k - \lambda_{\rho,k}^*, \big(Y_k^\top Y_k + \rho I\big)\lambda_k\rangle$$
$$= \langle\lambda_k - \lambda_{\rho,k}^*, (Y_k^\top Y_k - \nabla F(x_k)^\top\nabla F(x_k))\lambda_k\rangle + \langle\lambda_k - \lambda_{\rho,k}^*, \big(\nabla F(x_k)^\top\nabla F(x_k) + \rho I\big)\lambda_k\rangle$$
$$\geq -2C_y\|\lambda_k - \lambda_{\rho,k}^*\|\|Y_k - \nabla F(x_k)\| + \langle\lambda_k - \lambda_{\rho,k}^*, \big(\nabla F(x_k)^\top\nabla F(x_k) + \rho I\big)(\lambda_k - \lambda_{\rho,k}^*)\rangle$$
$$+ \langle\lambda_k - \lambda_{\rho,k}^*, \big(\nabla F(x_k)^\top\nabla F(x_k) + \rho I\big)\lambda_{\rho,k}^*\rangle$$
$$\geq -2C_y\|\lambda_k - \lambda_{\rho,k}^*\|\|Y_k - \nabla F(x_k)\| + \rho\|\lambda_k - \lambda_{\rho,k}^*\|^2$$
$$\geq -2C_y^2\rho^{-1}\|Y_k - \nabla F(x_k)\|^2 + \frac{\rho}{2}\|\lambda_k - \lambda_{\rho,k}^*\|^2 \tag{51}$$

where the second last inequality follows from the optimality condition that

$$\langle\lambda_k - \lambda_{\rho,k}^*, \big(\nabla F(x_k)^\top\nabla F(x_k) + \rho I\big)\lambda_{\rho,k}^*\rangle \geq 0,$$

and the last inequality in follows from the Young's inequality.

Plugging in (51) back to (50) gives

$$\|\lambda_{k+1} - \lambda_{\rho,k}^*\|^2 = (1 - \rho\gamma_k)\|\lambda_k - \lambda_{\rho,k}^*\|^2 + 4C_y^2\rho^{-1}\gamma_k\|Y_k - \nabla F(x_k)\|^2 + (C_y^2 + \rho)^2\gamma_k^2. \tag{52}$$

With Lemma 4, the second term in (49) can be bounded as

$$\langle\lambda_{k+1} - \lambda_{\rho,k}^*, \lambda_{\rho,k}^* - \lambda_{\rho,k+1}^*\rangle \leq L_{\lambda,F}\|\lambda_{k+1} - \lambda_{\rho,k}^*\|\|x_k - x_{k+1}\|$$
$$\leq L_{\lambda,F}C_y\alpha_k\|\lambda_{k+1} - \lambda_{\rho,k}^*\|$$
$$\leq \frac{\rho}{4}\gamma_k\|\lambda_{k+1} - \lambda_{\rho,k}^*\|^2 + L_{\lambda,F}^2 C_y^2\rho^{-1}\frac{\alpha_k^2}{\gamma_k}, \tag{53}$$

where the last inequality is due to Young's inequality. The last term in (49) is bounded as

$$\|\lambda_{\rho,k}^* - \lambda_{\rho,k+1}^*\|^2 \le L_{\lambda,F}^2 C_y^2 \alpha_k^2. \tag{54}$$

Substituting (52)–(54) into (49) yields

$$\|\lambda_{k+1} - \lambda_{\rho,k+1}^*\|^2 \le (1 - \frac{\rho}{2}\gamma_k)\|\lambda_k - \lambda_{\rho,k}^*\|^2 + 4C_y^2\rho^{-1}\gamma_k\|Y_k - \nabla F(x_k)\|^2 + (C_y^2 + \rho)^2\gamma_k^2$$

$$+ 2L_{\lambda,F}^2 C_y^2\rho^{-1}\frac{\alpha_k^2}{\gamma_k} + L_{\lambda,F}^2 C_y^2\alpha_k^2. \tag{55}$$

Suppose $\alpha_k$, $\beta_k$, and $\gamma_k$ are constants given $K$. Taking total expectation, rearranging and taking telescoping sum on both sides of the last inequality gives

$$\frac{1}{K}\sum_{k=1}^K \mathbb{E}\|\lambda_k - \lambda_{\rho,k}^*\|^2 = \mathcal{O}\left(\frac{1}{K\rho\gamma_k}\right) + \mathcal{O}\left(\frac{1}{\rho K}\sum_{k=1}^K \mathbb{E}\|Y_k - \nabla F(x_k)\|^2\right) + \mathcal{O}\left(\frac{\gamma_k}{\rho}\right)$$

$$+ \mathcal{O}\left(\frac{\alpha_k^2}{\gamma_k^2\rho^4}\right) + \mathcal{O}\left(\frac{\alpha_k^2}{\gamma_k\rho^3}\right) \tag{56}$$

where we have used $L_{\lambda,F} = \mathcal{O}(\frac{1}{\rho})$ from Lemma 4. Then, plugging in the choices $\alpha_k = \Theta(K^{-\frac{9}{10}})$, $\beta_k = \Theta(K^{-\frac{1}{2}})$, $\gamma_k = \Theta(K^{-\frac{2}{5}})$, $\rho = \Theta(K^{-\frac{1}{5}})$ and substituting from (48) in (56) gives

$$\frac{1}{K}\sum_{k=1}^K \mathbb{E}\|\lambda_k - \lambda_{\rho,k}^*\|^2 = \mathcal{O}\left(MK^{-\frac{3}{10}} + K^{-\frac{1}{5}}\right). \tag{57}$$

Since typically the number of objectives is very small compared to the number of iterations, under assumption $K = \mathcal{O}(M^{10})$, we get

$$\frac{1}{K}\sum_{k=1}^K \mathbb{E}\|\lambda_k - \lambda_{\rho,k}^*\|^2 = \mathcal{O}\left(K^{-\frac{1}{5}}\right). \tag{58}$$

Thus, from (36), (38), (48), and (58), we have

$$\frac{1}{K}\sum_{k=1}^K \mathbb{E}\|d(x_k) - Y_k\lambda_k\|^2 = \mathcal{O}\left(K^{-\frac{1}{5}}\right). \tag{59}$$

This completes the proof. $\qquad\square$

## G  PROOF OF THEOREM 1

*Proof.* We first have

$$\|x_{k+1} - x^*\|^2 = \|\Pi_{\mathcal{X}}(x_k - \alpha_k Y_k\lambda_k) - x^*\|^2$$

$$\le \|x_k - \alpha_k Y_k\lambda_k - x^*\|^2$$

$$= \|x_k - x^*\|^2 - 2\alpha_k\langle x_k - x^*, Y_k\lambda_k\rangle + \alpha_k^2\|Y_k\lambda_k\|^2$$

$$\le \|x_k - x^*\|^2 - 2\alpha_k\langle x_k - x^*, \nabla F(x_k)\lambda_k^*\rangle - 2\alpha_k\langle x_k - x^*, Y_k\lambda_k - \nabla F(x_k)\lambda_k^*\rangle + \alpha_k^2 C_y^2$$

$$\le \|x_k - x^*\|^2 - 2\alpha_k\lambda_k^* \cdot (F(x_k) - F(x^*)) - 2\alpha_k\langle x_k - x^*, Y_k\lambda_k - \nabla F(x_k)\lambda_k^*\rangle + \alpha_k^2 C_y^2 \tag{60}$$

where the last inequality uses the convexity of $f_m(x)(\forall m)$.

The third term in (60) can be bounded using the Cauchy–Schwarz inequality as

$$\langle x_k - x^*, Y_k\lambda_k - \nabla F(x_k)\lambda_k^*\rangle \ge -C_x(\|Y_k - \nabla F(x_k)\| + C_y\|\lambda_k - \lambda_{\rho,k}^*\| + \|\nabla F(x_k)\lambda_k^* - \nabla F(x_k)\lambda_{\rho,k}^*\|) \tag{61}$$

where $C_x$ is the upper bound of $\|x - x'\|$ for $x, x' \in \mathcal{X}$, which follows from Assumption 2.

Substituting the above inequality into (60) and rearranging gives

$$\lambda_k^*(F(x_k) - F(x^*)) \le \frac{1}{2\alpha_k}(\|x_k - x^*\|^2 - \|x_{k+1} - x^*\|^2) + C_x(\|Y_k - \nabla F(x_k)\| + C_y\|\lambda_k - \lambda_{\rho,k}^*\|$$

$$+ \|\nabla F(x_k)\lambda_k^* - \nabla F(x_k)\lambda_{\rho,k}^*\|) + \frac{C_y^2}{2}\alpha_k. \tag{62}$$

Taking telescope sum on the last inequality gives

$$\frac{1}{K}\sum_{k=1}^{K}\lambda_k^*(F(x_k) - F(x^*))$$

$$\le \frac{\|x_1 - x^*\|^2}{2K\alpha_k} + \frac{C_x}{K}\sum_{k=1}^{K}(\|Y_k - \nabla F(x_k)\| + C_y\|\lambda_k - \lambda_{\rho,k}^*\| + \|\nabla F(x_k)\lambda_k^* - \nabla F(x_k)\lambda_{\rho,k}^*\|) + \frac{C_y^2}{2}\alpha_k \tag{63}$$

which along with (38), (48), and (58) indicates $\frac{1}{K}\sum_{k=1}^{K}\mathbb{E}[\lambda_k^* \cdot (F(x_k) - F(x^*))] = \mathcal{O}(K^{-\frac{1}{10}})$ if we choose $\alpha_k = \Theta(K^{-\frac{9}{10}})$, $\beta_k = \Theta(K^{-\frac{1}{2}})$, $\gamma_k = \Theta(K^{-\frac{2}{5}})$, and $\rho = \Theta(K^{-\frac{1}{5}})$. $\qquad\square$

## H  PROOF OF THEOREM 2

Before we go into the main proof, we first show the following key lemma.

**Lemma 6.** *For any $x \in \mathcal{X}$ and $\lambda \in \Delta^M$, with $d(x) = \nabla F(x)\lambda^*(x)$, it holds that*

$$\langle d(x), \nabla F(x)\lambda \rangle \ge \|d(x)\|^2. \tag{64}$$

*Proof.* We write $d_\lambda(x) = \nabla F(x)\lambda$ in the following proof. Since $\Delta^M$ is a convex set, for any $\lambda' \in \Delta^M$, we have $\alpha(\lambda' - \lambda^*) + \lambda^* \in \Delta^M$ for any $\alpha \in [0, 1]$. Then by $d(x) = \arg\min_{\lambda \in \Delta^M}\|d_\lambda(x)\|^2$, we have

$$\|d(x)\|^2 \le \|\alpha(d_{\lambda'}(x) - d(x)) + d(x)\|^2. \tag{65}$$

Expanding the right hand side of the inequality gives

$$\alpha^2\|d_{\lambda'}(x) - d(x)\|^2 + \alpha\langle d(x), d_{\lambda'}(x) - d(x)\rangle \ge 0. \tag{66}$$

Since this needs to hold for $\alpha$ arbitrarily close to 0, we have

$$\langle d(x), d_{\lambda'}(x) - d(x)\rangle \ge 0, \quad \forall \lambda' \in \Delta^M \tag{67}$$

which indicates the result inequality by rearranging. $\qquad\square$

Now we can prove Theorem 2.

*Proof.* By the $L_{m,1}$-smoothness of $f_m$, we have for any $m$,

$$f_m(x_{k+1}) \le f_m(x_k) + \alpha_k\langle\nabla f_m(x_k), -Y_k\lambda_k\rangle + \frac{L_{m,1}}{2}\|x_{k+1} - x_k\|^2$$

$$\le f_m(x_k) + \alpha_k\langle\nabla f_m(x_k), -Y_k\lambda_k\rangle + \frac{L_{m,1}}{2}C_y^2\alpha_k^2. \tag{68}$$

The second term in the last inequality can be bounded as

$$\langle\nabla f_m(x_k), -Y_k\lambda_k\rangle$$
$$= \langle\nabla f_m(x_k), \nabla F(x_k)\lambda_k^* - Y_k\lambda_k\rangle + \langle\nabla f_m(x_k), -\nabla F(x_k)\lambda_k^*\rangle$$
$$\le L_m(\|Y_k - \nabla F(x_k)\| + C_y\|\lambda_k - \lambda_{\rho,k}^*\| + \|\nabla F(x_k)\lambda_k^* - \nabla F(x_k)\lambda_{\rho,k}^*\|) + \langle\nabla f_m(x_k), -\nabla F(x_k)\lambda_k^*\rangle$$
$$\le L_m(\|Y_k - \nabla F(x_k)\| + C_y\|\lambda_k - \lambda_{\rho,k}^*\| + \|\nabla F(x_k)\lambda_k^* - \nabla F(x_k)\lambda_{\rho,k}^*\|) - \|\nabla F(x_k)\lambda_k^*\|^2, \tag{69}$$

where the last inequality follows from Lemma 6 by letting $\nabla F(x_k)\lambda = \nabla f_m(x_k)$, and the first inequality follows from

$$
\begin{aligned}
&\langle \nabla f_m(x_k), \nabla F(x_k)\lambda_k^* - Y_k\lambda_k \rangle \\
&\leq L_m \|\nabla F(x_k)\lambda_k^* - Y_k\lambda_k\| \\
&\leq L_m \|\nabla F(x_k)\lambda_k - Y_k\lambda_k + \nabla F(x_k)\lambda_{\rho,k}^* - \nabla F(x_k)\lambda_k + \nabla F(x_k)\lambda_k^* - \nabla F(x_k)\lambda_{\rho,k}^*\| \\
&\leq L_m \big( \|Y_k - \nabla F(x_k)\| + C_y\|\lambda_k - \lambda_{\rho,k}^*\| + \|\nabla F(x_k)\lambda_k^* - \nabla F(x_k)\lambda_{\rho,k}^*\| \big).
\end{aligned}
\tag{70}
$$

Plugging (69) into (68), taking expectation on both sides and rearranging yields

$$
\begin{aligned}
\alpha_k \mathbb{E}\|\nabla F(x_k)\lambda_k^*\|^2 \leq{}& \mathbb{E}[f_m(x_k) - f_m(x_{k+1})] + L_m\alpha_k\big(\mathbb{E}\|Y_k - \nabla F(x_k)\| + C_y\mathbb{E}\|\lambda_k - \lambda_{\rho,k}^*\| \\
&+ \|\nabla F(x_k)\lambda_k^* - \nabla F(x_k)\lambda_{\rho,k}^*\|\big) + \frac{L_{m,1}}{2}C_y^2\alpha_k^2.
\end{aligned}
\tag{71}
$$

Taking telescope sum on both sides of the last inequality gives

$$
\begin{aligned}
\frac{1}{K}\sum_{k=1}^{K}\mathbb{E}\|\nabla F(x_k)\lambda_k^*\|^2 \leq{}& \frac{1}{\alpha_k K}(f_m(x_1) - \inf f_m(x)) + L_m\frac{1}{K}\sum_{k=1}^{K}\big(\mathbb{E}\|Y_k - \nabla F(x_k)\| + C_y\mathbb{E}\|\lambda_k - \lambda_{\rho,k}^*\| \\
&+ \|\nabla F(x_k)\lambda_k^* - \nabla F(x_k)\lambda_{\rho,k}^*\|\big) + \frac{L_{m,1}}{2}C_y^2\alpha_k
\end{aligned}
\tag{72}
$$

which along with (38), (48), and (58) indicates $\frac{1}{K}\sum_{k=1}^{K}\mathbb{E}\|\nabla F(x_k)\lambda_k^*\|^2 = \mathcal{O}(K^{-\frac{1}{10}})$ if we choose $\alpha_k = \Theta(K^{-\frac{9}{10}})$, $\beta_k = \Theta(K^{-\frac{1}{2}})$, $\gamma_k = \Theta(K^{-\frac{2}{5}})$, and $\rho = \Theta(K^{-\frac{1}{5}})$. $\qquad\square$

## I   PROOF OF THEOREM 3

*Proof.* Recall from (68), by the $L_{m,1}$-smoothness of $f_m$, we have for any $m$,

$$
f_m(x_{k+1}) \leq f_m(x_k) + \alpha_k\langle \nabla f_m(x_k), -Y_k\lambda_k \rangle + \frac{L_{m,1}}{2}C_y^2\alpha_k^2.
\tag{73}
$$

Multiplying both sides by $\lambda_k^m$ and summing over all $m \in [M]$, we obtain

$$
F(x_{k+1})\lambda_k \leq F(x_k)\lambda_k + \alpha_k\langle \nabla F(x_k)\lambda_k, -Y_k\lambda_k \rangle + \frac{\bar{L}_1}{2}C_y^2\alpha_k^2,
\tag{74}
$$

where we have used $\lambda_k := (\lambda_k^1, \lambda_k^2, \ldots, \lambda_k^m, \ldots, \lambda_k^M)^\top$ and $\bar{L}_1 = \max_m L_{m,1}$. We can bound the second term of (74) as

$$
\begin{aligned}
\langle \nabla F(x_k)\lambda_k, -Y_k\lambda_k \rangle &= \langle \nabla F(x_k), -\nabla F(x_k) + \nabla F(x_k) - Y_k \rangle \\
&\leq -\|\nabla F(x_k)\lambda_k\|^2 + \frac{1}{2}\|\nabla F(x_k)\lambda_k\|^2 + \frac{1}{2}\|\nabla F(x_k) - Y_k\|^2 \\
&= -\frac{1}{2}\|\nabla F(x_k)\lambda_k\|^2 + \frac{1}{2}\|\nabla F(x_k) - Y_k\|^2,
\end{aligned}
\tag{75}
$$

where the first inequality is due to Cauchy-Schwartz and Young's inequalities. Substituting (75) in (74) and rearranging, we have

$$
\frac{\alpha_k}{2}\|\nabla F(x_k)\lambda_k\|^2 \leq F(x_k)\lambda_k - F(x_{k+1})\lambda_k + \frac{\alpha_k}{2}\|\nabla F(x_k) - Y_k\|^2 + \frac{L\bar{L}_1}{2}\alpha_k^2 C_y^2.
\tag{76}
$$

Given $K$, let $\alpha_k, \beta_k$ and $\gamma_k$ be constants for any $k \in [K]$. We then take total expectation on both sides and sum over iterations to obtain

$$
\frac{\alpha_k}{2}\sum_{k=1}^{K}\mathbb{E}\|\nabla F(x_k)\lambda_k\|^2 \leq \sum_{k=1}^{K}\mathbb{E}\left[F(x_k)\lambda_k - F(x_{k+1})\lambda_k\right] + \frac{\alpha_k}{2}\sum_{k=1}^{K}\mathbb{E}\|\nabla F(x_k) - Y_k\|^2 + \frac{\bar{L}_1}{2}\alpha_k^2 K C_y^2.
\tag{77}
$$

We bound the first term on the right-hand side of the inequality (77) as

$$
\begin{aligned}
\sum_{k=1}^{K} \mathbb{E}\left[F(x_k)\lambda_k - F(x_{k+1})\lambda_k\right] &= \mathbb{E}\left[\sum_{k=1}^{K-1} F(x_{k+1})(\lambda_{k+1} - \lambda_k) + F(x_1)\lambda_1 - F(x_{K+1})\lambda_k\right] \\
&\leq \mathbb{E}\left[\sum_{k=1}^{K-1} \|F(x_{k+1})\|\|\lambda_{k+1} - \lambda_k\| + \|F(x_1)\|\|\lambda_1\| + \|F(x_{K+1})\|\|\lambda_k\|\right] \\
&\leq F\sum_{k=1}^{K-1} \|\gamma_k Y_k^{\top} Y_k \lambda_k\| + 2F \\
&\leq FC_y^2(K-1)\gamma_k + 2F,
\end{aligned}
\tag{78}
$$

where the first inequality is due to Cauchy-Schwartz, the second inequality is due to the bounds on $F(x_k)$, $\lambda_k$ and we have used the update for $\lambda_k$ for all $k \in [K]$ with $\rho = 0$, and third inequality is due to the bound on $Y_k$ for all $k \in [K]$. Substituting (78) in (77) and dividing both sides by $\frac{\alpha_k K}{2}$, we have

$$
\frac{1}{K}\sum_{k=1}^{K} \mathbb{E}\|\nabla F(x_k)\lambda_k\|^2 \leq 2FC_y^2 \frac{(K-1)}{K}\frac{\gamma_k}{\alpha_k} + 4F\frac{1}{\alpha_k K} + \frac{1}{K}\sum_{k=1}^{K} \mathbb{E}\|\nabla F(x_k) - Y_k\|^2 + \bar{L}_1 \alpha_k C_y^2
\tag{79}
$$

which, along with (46) and choosing $\alpha_k = \Theta(K^{-\frac{3}{5}})$, $\beta_k = \Theta(K^{-\frac{2}{5}})$, and $\gamma_k = \Theta(K^{-1})$, we obtain

$$
\frac{1}{K}\sum_{k=1}^{K} \mathbb{E}\|\nabla F(x_k)\lambda_k\|^2 = \mathcal{O}(MK^{-\frac{2}{5}}).
\tag{80}
$$

The result then follows by observing that for any $k \in [K]$, we have

$$
\|\nabla F(x_k)\lambda_k\|^2 \geq \min_{\lambda}\|\nabla F(x_k)\lambda\|^2 = \|\nabla F(x_k)\lambda_k^*\|^2.
\tag{81}
$$

$\square$

## J    IMPROVED CONVERGENCE RATE WITH MODIFIED ASSUMPTIONS

In this section we state and prove Theorem 4, which improves upon the results presented in Theorem 3, with modified assumptions. We will first state the modified assumptions.

**Assumption 4.** *For any $m$, there exist constants $c_m, \sigma_m$ such that $\frac{1}{K}\sum_{k=1}^{K} \mathbb{E}\|\mathbb{E}[h_{k,m}|\mathcal{F}_k] - \nabla f_m(x_k)\|^2 \leq c_m' \beta_K$ and $\mathbb{E}[\|h_{k,m} - \mathbb{E}[h_{k,m}|\mathcal{F}_k]\|^2|\mathcal{F}_k] \leq \sigma_m^2$ for any $k$.*

Similar to Assumption 3, Assumption 4 requires the stochastic gradient $h_{k,m}$ almost unbiased and has bounded variance, and this is also weaker than the standard unbiased stochastic gradient assumption. Furthermore, Assumption 4 can be satisfied by running multiple nested updates, which require additional lower-level samples. With this assumption, we present the following improved result.

**Theorem 4.** *Consider the sequences generated by Algorithm 1. Furthermore assume there exists a constant $F > 0$ such that for all $k \in [K]$, $\|F(x_k)\| \leq F$. Then, under Assumptions 1 and 4, if we choose step sizes $\alpha_k = \Theta(K^{-\frac{1}{2}})$, $\beta_k = \Theta(K^{-\frac{1}{2}})$, $\gamma_k = \Theta(K^{-\frac{3}{4}})$, and $\rho = 0$, it holds that*

$$
\frac{1}{K}\sum_{k=1}^{K} \mathbb{E}\left[\|\nabla F(x_k)\lambda^*(x_k)\|^2\right] = \mathcal{O}\left(MK^{-\frac{1}{2}}\right).
\tag{82}
$$

*Proof.* **Convergence of $Y_k$.** We begin the proof by revisiting the convergence analysis on $Y_k$ in the proof of Theorem 3, under the assumptions considered in Theorem 4. For convenience, we restate (39) here as

$$
\begin{aligned}
\mathbb{E}_k\|y_{k+1,m} - \nabla f_m(x_{k+1})\|^2 = \mathbb{E}_k\|y_{k+1,m} - \nabla f_m(x_k)\|^2 &+ 2\mathbb{E}_k\langle y_{k+1,m} - \nabla f_m(x_k), \nabla f_m(x_k) - \nabla f_m(x_{k+1})\rangle \\
&+ \mathbb{E}_k\|\nabla f_m(x_k) - \nabla f_m(x_{k+1})\|^2.
\end{aligned}
\tag{83}
$$

We bound the first term in (83) similar to that in (42), as

$$
\mathbb{E}_k \|y_{k+1,m} - \nabla f_m(x_k)\|^2 \leq \left(1 - \frac{1}{2}\beta_k\right)\|y_{k,m} - \nabla f_m(x_k)\|^2 + \beta_k \|\nabla f_m(x_k) - \mathbb{E}_k[h_{k,m}]\|^2
$$
$$
+ 3\beta_k^2 \|\nabla f_m(x_k) - \mathbb{E}_k[h_{k,m}]\|^2 + 3\sigma_m^2 \beta_k^2. \tag{84}
$$

The second term in (83) can be bounded as

$$
\langle y_{k+1,m} - \nabla f_m(x_k), \nabla f_m(x_k) - \nabla f_m(x_{k+1})\rangle
$$
$$
\leq \|y_{k+1,m} - \nabla f_m(x_k)\|\|\nabla f_m(x_k) - \nabla f_m(x_{k+1})\|
$$
$$
\leq L_{m,1}\|y_{k+1,m} - \nabla f_m(x_k)\|\|x_{k+1} - x_k\|
$$
$$
\leq L_{m,1}\alpha_k\|y_{k+1,m} - \nabla f_m(x_k)\|\|Y_k\lambda_k\|
$$
$$
\leq \frac{1}{8}\beta_k\|y_{k+1,m} - \nabla f_m(x_k)\|^2 + 2L_{m,1}^2\frac{\alpha_k^2}{\beta_k}\|Y_k\lambda_k\|^2
$$
$$
\leq \frac{1}{8}\beta_k\|y_{k+1,m} - \nabla f_m(x_k)\|^2 + 2\bar{L}_1^2\frac{\alpha_k^2}{\beta_k}\|Y_k\lambda_k\|^2 \tag{85}
$$

where the third inequality is due to the $x_k$ update, the second last inequality follows from young's inequality, and the last inequality follows from the definition $\bar{L}_1 = \max_m L_{m,1}$.

The last term in (83) can be bounded as

$$
\|\nabla f_m(x_k) - \nabla f_m(x_{k+1})\|^2 \leq \bar{L}_1^2 \|x_{k+1} - x_k\|^2 \leq \bar{L}_1^2 \alpha_k^2 \|Y_k\lambda_k\|^2. \tag{86}
$$

Collecting the upper bounds in (84)–(86) and substituting them into (83) gives

$$
\mathbb{E}_k\|y_{k+1,m} - \nabla f_m(x_{k+1})\|^2 \leq (1 - \frac{1}{4}\beta_k)\|y_{k,m} - \nabla f_m(x_k)\|^2 + (\beta_k + 3\beta_k^2)\|\nabla f_m(x_k) - \mathbb{E}_k[h_{k,m}]\|^2
$$
$$
+ 3\sigma_m^2\beta_k^2 + \left(\bar{L}_1^2\alpha_k^2 + 4\bar{L}_1^2\frac{\alpha_k^2}{\beta_k}\right)\|Y_k\lambda_k\|^2. \tag{87}
$$

For all $k$, let $\alpha_k = \alpha_K$, $\beta_k = \beta_K$, and $\gamma_k = \gamma_K$ be constants given $K$. Then, taking total expectation and then telescoping both sides of (87) gives

$$
\frac{1}{K}\sum_{k=1}^{K}\mathbb{E}\|y_{k,m} - \nabla f_m(x_k)\|^2 = \mathcal{O}\left(\frac{1}{K\beta_K}\right) + \mathcal{O}\left(\frac{1}{K}\sum_{k=1}^{K}\mathbb{E}\|\nabla f_m(x_k) - \mathbb{E}_k[h_{k,m}]\|^2\right)
$$
$$
+ \mathcal{O}(\beta_K) + \left(\bar{L}_1^2\frac{\alpha_K^2}{\beta_K} + 4\bar{L}_1^2\frac{\alpha_K^2}{\beta_K^2}\right)\frac{1}{K}\sum_{k=1}^{K}\mathbb{E}\|Y_k\lambda_k\|^2. \tag{88}
$$

Summing the last inequality over all objectives $m \in [M]$, and using Assumption 4, we obtain

$$
\frac{1}{K}\sum_{k=1}^{K}\mathbb{E}\|Y_k - \nabla F(x_k)\|^2 = \mathcal{O}\left(\frac{M}{K\beta_K}\right) + \mathcal{O}(M\beta_K) + \left(M\bar{L}_1^2\alpha_K^2 + 4M\bar{L}_1^2\frac{\alpha_K^2}{\beta_K}\right)\frac{1}{K}\sum_{k=1}^{K}\mathbb{E}\|Y_k\lambda_k\|^2, \tag{89}
$$

where we have used $\sum_{m=1}^{M}\|y_{k,m} - \nabla f_m(x_k)\|^2 \geq \|Y_k - \nabla F(x_k)\|^2$. Then with the decomposition

$$
\|Y_k\lambda_k\|^2 \leq 2\|Y_k - \nabla F(x_k)\|^2 + 2\|\nabla F(x_k)\lambda_k\|^2, \tag{90}
$$

we can arrive at

$$
\frac{1}{K}\sum_{k=1}^{K}\mathbb{E}\|Y_k - \nabla F(x_k)\|^2 = \mathcal{O}\left(\frac{M}{K\beta_K}\right) + \mathcal{O}(M\beta_K) + \left(2M\bar{L}_1^2\alpha_K^2 + 8M\bar{L}_1^2\frac{\alpha_K^2}{\beta_K}\right)\frac{1}{K}\sum_{k=1}^{K}\mathbb{E}\|\nabla F(x_k)\lambda_k\|^2
$$
$$
+ \left(2M\bar{L}_1^2\alpha^2 + 8M\bar{L}_1^2\frac{\alpha_K^2}{\beta_K}\right)\frac{1}{K}\sum_{k=1}^{K}\mathbb{E}\|Y_k - \nabla F(x_k)\|^2. \tag{91}
$$

Now, note that with the choice of stepsize $\alpha_K \leq \frac{1}{4\bar{L}\sqrt{M}}\beta_K$, $0 < \beta_K < 1$, and $\alpha_K$ and $\beta_K$ are on the same time scale, there exist some constant $0 < C_1 < 1$ and valid choice of $\beta_K$ such that the

following inequality holds

$$1 - 2M\bar{L}_1^2\alpha_K^2 - 8M\bar{L}_1^2\frac{\alpha_K^2}{\beta_K} \geq C_1. \tag{92}$$

An example for a constant that satisfy the last inequality for aforementioned choice of $\alpha_K$ and $\beta_K$ is $C_1 = \frac{1}{4}$. Then, from (91) and (92), we can arrive at

$$\frac{1}{K}\sum_{k=1}^{K}\mathbb{E}\|Y_k - \nabla F(x_k)\|^2 = \mathcal{O}\left(\frac{M}{K\beta_K}\right) + \mathcal{O}(M\beta_K) + \left(\frac{2M\bar{L}_1^2}{C_1}\alpha_K^2 + \frac{8M\bar{L}_1^2}{C_1}\frac{\alpha_K^2}{\beta_K}\right)\frac{1}{K}\sum_{k=1}^{K}\mathbb{E}\|\nabla F(x_k)\lambda_k\|^2 \tag{93}$$

Next, we analyse the $x_k$ sequence. For this purpose, we follow along similar lines as in the proof of Theorem 3. Accordingly, from (68), we have for any $m$,

$$f_m(x_{k+1}) \leq f_m(x_k) + \alpha_k\langle\nabla f_m(x_k), -Y_k\lambda_k\rangle + \frac{L_{m,1}}{2}C_y^2\alpha_k^2. \tag{94}$$

Multiplying both sides by $\lambda_k^m$ and summing over all $m \in [M]$, we obtain

$$F(x_{k+1})\lambda_k \leq F(x_k)\lambda_k + \alpha_k\langle\nabla F(x_k)\lambda_k, -Y_k\lambda_k\rangle + \frac{\bar{L}_1}{2}C_y^2\alpha_k^2, \tag{95}$$

where we have used $\lambda_k := (\lambda_k^1, \lambda_k^2, \ldots, \lambda_k^m, \ldots, \lambda_k^M)^\top$ and $\bar{L}_1 = \max_m L_{m,1}$. We can bound the second term of (95) as

$$\begin{aligned}
\langle\nabla F(x_k)\lambda_k, -Y_k\lambda_k\rangle &= \langle\nabla F(x_k)\lambda_k, -\nabla F(x_k)\lambda_k + \nabla F(x_k)\lambda_k - Y_k\lambda_k\rangle \\
&\leq -\|\nabla F(x_k)\lambda_k\|^2 + \langle\nabla F(x_k)\lambda_k, \nabla F(x_k)\lambda_k - Y_k\lambda_k\rangle \\
&\leq -\|\nabla F(x_k)\lambda_k\|^2 + \frac{1}{2}\|\nabla F(x_k)\lambda_k\|^2 + \frac{1}{2}\|\nabla F(x_k) - Y_k\lambda_k\|^2 \\
&\leq -\frac{1}{2}\|\nabla F(x_k)\lambda_k\|^2 + \frac{1}{2}\|\nabla F(x_k) - Y_k\|^2,
\end{aligned} \tag{96}$$

where the second inequality is due to Cauchy-Schwartz and Young's inequalities, and the last inequality is due to bound on $\lambda_k$. Substituting (96) in (95) and rearranging, we have

$$\frac{\alpha_k}{2}\|\nabla F(x_k)\lambda_k\|^2 \leq F(x_k)\lambda_k - F(x_{k+1})\lambda_k + \frac{\alpha_k}{2}\|\nabla F(x_k) - Y_k\|^2 + \frac{\bar{L}_1}{2}\alpha^2 C_y^2. \tag{97}$$

For all $k$, let $\alpha_k = \alpha_K$, $\beta_k = \beta_K$, and $\gamma_k = \gamma_K$ be constants given $K$. We then take total expectation on both sides and sum over iterations to obtain

$$\frac{\alpha_K}{2}\sum_{k=1}^{K}\mathbb{E}\|\nabla F(x_k)\lambda_k\|^2 \leq \sum_{k=1}^{K}\mathbb{E}\left[F(x_k)\lambda_k - F(x_{k+1})\lambda_k\right] + \frac{\alpha_K}{2}\sum_{k=1}^{K}\mathbb{E}\|\nabla F(x_k) - Y_k\|^2 + \frac{\bar{L}_1}{2}\alpha_K^2KC_y^2. \tag{98}$$

We bound the first term on the right-hand side of the inequality (98) as

$$\begin{aligned}
\sum_{k=1}^{K}\mathbb{E}\left[F(x_k)\lambda_k - F(x_{k+1})\lambda_k\right] &= \mathbb{E}\left[\sum_{k=1}^{K-1}F(x_{k+1})(\lambda_{k+1} - \lambda_k) + F(x_1)\lambda_1 - F(x_{K+1})\lambda_k\right] \\
&\leq \mathbb{E}\left[\sum_{k=1}^{K-1}\|F(x_{k+1})\|\|\lambda_{k+1} - \lambda_k\| + \|F(x_1)\|\|\lambda_1\| + \|F(x_{K+1})\|\|\lambda_k\|\right] \\
&\leq F\sum_{k=1}^{K-1}\|\gamma_KY_k^\top Y_k\lambda_k\| + 2F \\
&\leq FC_y\sum_{k=1}^{K}\gamma_K\|Y_k\lambda_k\| + 2F,
\end{aligned} \tag{99}$$

where the first inequality is due to Cauchy-Schwartz, the second inequality is due to the bounds on $F(x_k)$, $\lambda_k$ and we have used the update for $\lambda_k$ for all $k \in [K]$ with $\rho = 0$, and third inequality is due to the bound on $Y_k$ for all $k \in [K]$.

Substituting (99) in (98) and dividing both sides by $\frac{\alpha_K K}{2}$, we have

$$\frac{1}{K}\sum_{k=1}^{K}\mathbb{E}\|\nabla F(x_k)\lambda_k\|^2$$

$$\leq 2FC_y\frac{1}{K}\frac{\gamma_K}{\alpha_K}\sum_{k=1}^{K}\mathbb{E}\|Y_k\lambda_k\| + 4F\frac{1}{\alpha_K K} + \frac{1}{K}\sum_{k=1}^{K}\mathbb{E}\|\nabla F(x_k) - Y_k\|^2 + \bar{L}_1\alpha_K C_y^2$$

$$\leq 2F^2C_y^2\frac{\gamma_K^2}{\alpha_K^2} + \frac{1}{4K}\sum_{k=1}^{K}\mathbb{E}\|Y_k\lambda_k\|^2 + 4F\frac{1}{\alpha_K K} + \frac{1}{K}\sum_{k=1}^{K}\mathbb{E}\|\nabla F(x_k) - Y_k\|^2 + \bar{L}_1\alpha_K C_y^2$$

$$\leq 2F^2C_y^2\frac{\gamma_K^2}{\alpha_K^2} + \frac{3}{2K}\sum_{k=1}^{K}\mathbb{E}\|\nabla F(x_k) - Y_k\|^2 + \frac{1}{2K}\sum_{k=1}^{K}\mathbb{E}\|\nabla F(x_k)\lambda_k\|^2 + 4F\frac{1}{\alpha_K K} + \bar{L}_1\alpha_K C_y^2$$

$$= \mathcal{O}\left(\frac{\gamma_K^2}{\alpha_K^2}\right) + \mathcal{O}\left(\frac{M}{K\beta_K}\right) + \mathcal{O}(M\beta_K) + \mathcal{O}\left(\frac{1}{\alpha_K K}\right) + \mathcal{O}(\alpha_K)$$

$$+ \left(\frac{1}{2} + \frac{3M\bar{L}_1^2}{C_1}\alpha_K^2 + \frac{12M\bar{L}_1^2}{C_1}\frac{\alpha_K^2}{\beta_K}\right)\frac{1}{K}\sum_{k=1}^{K}\mathbb{E}\|\nabla F(x_k)\lambda_k\|^2 \tag{100}$$

where the last equality is by substituting from (93). Now, with a similar argument that we made in (92), given some $C_1$, there exist some constant $0 < C_2 < 1$ such that

$$\frac{1}{2} - \frac{3M\bar{L}_1^2}{C_1}\alpha_K^2 - \frac{12M\bar{L}_1^2}{C_1}\frac{\alpha_K^2}{\beta_K} \geq C_2. \tag{101}$$

Feasible choices of $C_1, C_2$ would be $C_1 = C_2 = \frac{1}{4}$. Then, we can have

$$\frac{1}{K}\sum_{k=1}^{K}\mathbb{E}\|\nabla F(x_k)\lambda_k\|^2 = \mathcal{O}\left(\frac{\gamma_K^2}{\alpha_K^2}\right) + \mathcal{O}\left(\frac{M}{K\beta_K}\right) + \mathcal{O}(M\beta_K) + \mathcal{O}\left(\frac{1}{\alpha_K K}\right) + \mathcal{O}(\alpha_K). \tag{102}$$

By choosing $\alpha_K = \Theta(K^{-\frac{1}{2}})$, $\beta_K = \Theta(K^{-\frac{1}{2}})$, $\gamma_K = \Theta(K^{-\frac{3}{4}})$, we arrive at

$$\frac{1}{K}\sum_{k=1}^{K}\mathbb{E}\|\nabla F(x_k)\lambda_k\|^2 = \mathcal{O}(MK^{-\frac{1}{2}}). \tag{103}$$

The result then follows by observing that for any $k \in [K]$, we have

$$\|\nabla F(x_k)\lambda_k\|^2 \geq \min_{\lambda}\|\nabla F(x_k)\lambda\|^2 = \|\nabla F(x_k)\lambda_k^*\|^2. \tag{104}$$

$\square$

## K  DETAILS OF EXPERIMENTS

In this section, we describe the omitted details of experiments in the main paper.

### K.1  TOY EXAMPLE

To show the advantages of our algorithm, we use a toy example similar to (Liu et al., 2021a). The example consists of optimizing two objectives $f_1(x)$ and $f_2(x)$ with $x = (x_1, x_2)^\top \in \mathbb{R}^2$, given by

$$f_1(x) = p_1(x)q_1(x) + p_2(x)r_1(x); \quad f_2(x) = p_1(x)q_2(x) + p_2(x)r_2(x)$$

where we define

$q_1(x) = \log\left(|0.5\max\left(-x_1 - 7\right) - \tanh(-x_2)|, 0.000005\right) + 6,$

$q_2(x) = \log\left(|0.5\max\left(-x_1 - 7\right) - \tanh(-x_2) + 2|, 0.000005\right) + 6,$

$r_1(x) = \left((-x_1 + 7)^2 + 0.1\left(-x_1 - 8\right)^2\right)/10 - 20, \quad r_2(x) = \left((-x_1 - 7)^2 + 0.1\left(-x_1 - 8\right)^2\right)/10 - 20,$

$p_1(x) = \max\left(\tanh\left(0.5x_2\right), 0\right), \quad p_2(x) = \max\left(\tanh\left(-0.5x_2\right), 0\right).$

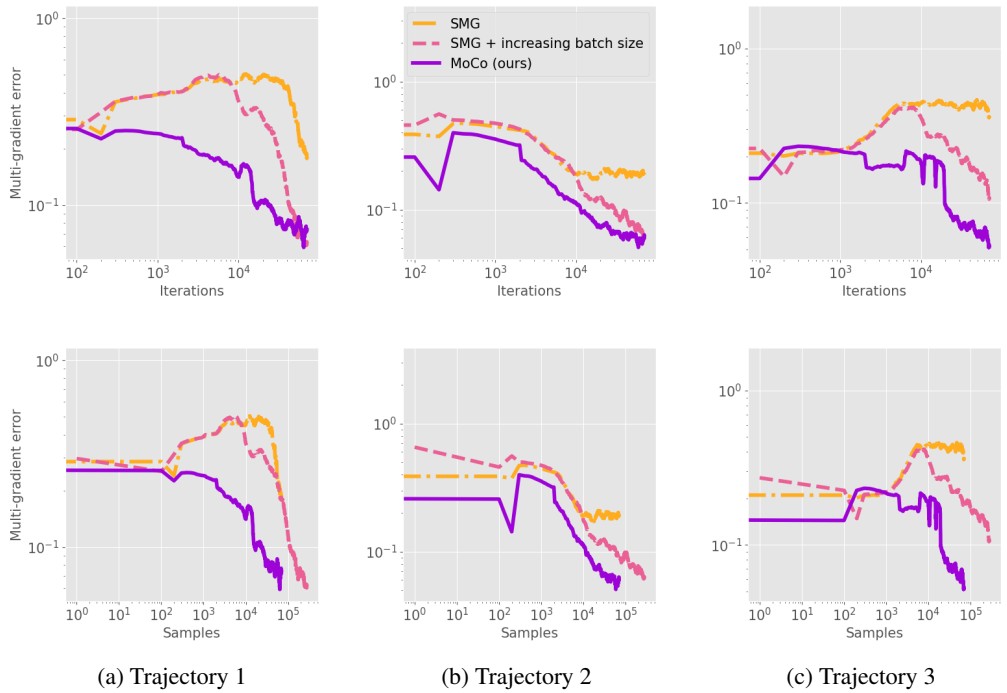

(a) Trajectory 1          (b) Trajectory 2          (c) Trajectory 3

Figure 3: Comparison of multi-gradient error

**Generation of Figure 1.** For generating the trajectories in Figure 1 we use 3 initializations

$$x_0 \in \{(-8.5, 7.5), (-8.5, 5), (10, -8)\},$$

and run each algorithm for 70000 iterations. For all the algorithms, we use the initial learning rate of 0.001, exponentially decaying at a rate of 0.05. In this example for MoCo, we use $\beta_k = 5/k^{0.5}$, where $k$ is the number of iterations.

**Generation of Figure 2.** For the comparison of MOO algorithms in objective space depicted in Figure 2, we use 5 initializations

$$x_0 \in \{(-8.5, 7.5), (-8.5, 5), (10, -8), (0, 0), (9, 9)\}.$$

The optimization configurations for each algorithm is similar to that of the aforementioned trajectory example, except with initial learning rate of 0.0025.

**Comparison with SMG with growing batch size.** For the comparison of the multi-gradient bias among SMG, SMG with increasingly large batch size, and MoCo, we use the norm of the error of the stochastic multi-gradient calculated using the three trajectories randomly initialized from $x_0 \in \{(-8.5, 7.5), (-8.5, 5), (10, -8)\}$. For calculating the bias of the multi-gradient, we compute the multi-gradient using 10 sets of gradient samples at each point of the trajectory, take the average and record the norm of the difference between the computed average and true multi-gradient. All three methods are run for 70000 iterations, and follow the same optimization configuration used for Figure 2. For SMG with increasing batch size, we increase the number of samples in the minibatch used for estimating the gradient by one every 10000 iterations. We report the bias of the multi-gradient with respect to the number of iterations and also number of samples in Figure 3. In the figure, Trajectories 1, 2, and 3 correspond to initializations $(-8.5, 7.5)$, $(-8.5, 5)$, and $(10, -8)$, respectively. It can be seen that our method performs comparable to SMG with increasing batch size, but with fewer samples. Furthermore, SMG has non-decaying bias in some trajectories.

### K.2   SUPERVISED LEARNING

In this section we provide additional details and experiments on Cityscapes and NYU-v2 datasets, and also provide experiment results on two additional datasets Office-31 and Office-home. Each experiment consists of solving multiple supervised learning problems related to each dataset. We

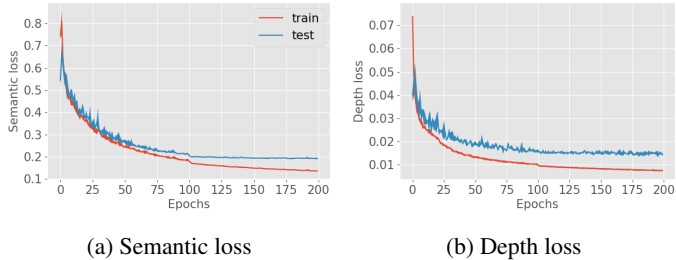

(a) Semantic loss

(b) Depth loss

Figure 4: Training and test loss for the Cityscapes tasks

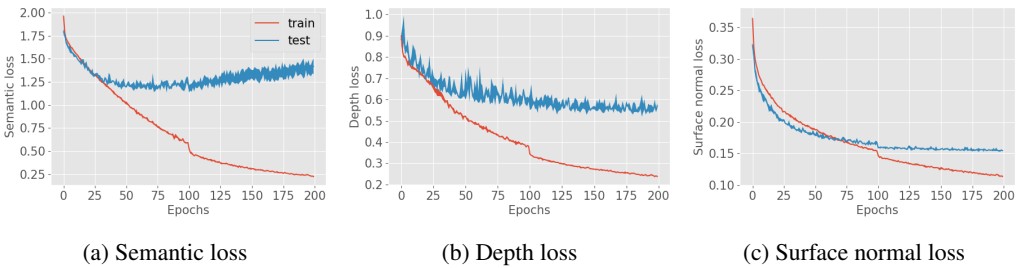

(a) Semantic loss

(b) Depth loss

(c) Surface normal loss

Figure 5: Training and test loss for the NYU-v2 tasks

consider each such task as an objective, which then can be simultaneously optimized by a gradient based MOO algorithm. We describe details on formulating each task as an objective next.

**Problem formulation.** First we look into the problem formulation of NYU-v2 and Cityscapes experiments. For NYU-v2 dataset, 3 tasks are involved: pixel-wise 13-class classification, pixel-wise depth estimation, and pixel-wise surface normal estimation. In Cityscapes experiments, there are 2 tasks involved: pixel-wise 7-class classification and pixel-wise depth estimation. The following problem formulation applies for both NYU-v2 and Cityscapes tasks, excpet for the surface normal estimation task which only relates to NYU-v2 dataset. Let the set of images be $\{U\}_{i=1}^{N}$, pixel-wise class labels be $\{T_1\}_{i=1}^{N}$, pixel-wise depth values be $\{T_2\}_{i=1}^{N}$, and pixel-wise surface normal values be $\{T_3\}_{i=1}^{N}$, where $N$ is the number of training data samples. Let $x$ be the model that we train to perform all the tasks simultaneously. Let the image dimension be $P \times Q$. We will use $T_m$ for ground truth and $\hat{T}_m$ for the corresponding prediction by the model $x$, where $m \in [M]$ and $M$ is the number of tasks. We can now formulate the corresponding objective for each task, as given in (Liu et al., 2019). The objective for pixel-wise classification is pixel-wise cross-entropy, which is given as

$$f_1(x) = -\frac{1}{NPQ} \sum_{i,p,q} T_{1,i}(p,q) \log \hat{T}_{1,i}(p,q)$$

where $i \in [N]$, $p \in [P]$, and $q \in [Q]$. Similarly, we can have the objectives for pixel-wise depth estimation and surface normal estimation, respectively, as

$$f_2(x) = \frac{1}{NPQ} \sum_{i,p,q} |T_{1,i}(p,q) - \hat{T}_{1,i}(p,q)| \quad \text{and} \quad f_3(x) = \frac{1}{NPQ} \sum_{i,p,q} T_{1,i}(p,q) \cdot \hat{T}_{1,i}(p,q),$$

where $\cdot$ is the elementwise dot product. With these objectives, we can formulate the problem (1) for Cityscapes and NYU-v2 tasks, with $f_3$ only used in the latter.

Similar to NYU-v2 and Cityscapes experiments, we can also formulate the supervised learning tasks on Office-31 and Office-home MTL as an instance of problem (1).

**Cityscapes dataset.** For implementing evaluation Cityscapes dataset, we follow the experiment set up used in (Liu et al., 2021a). All the MTL algorithms considered are trained using a SegNet (Badrinarayanan et al., 2017) model with attention mechanism MTAN (Liu et al., 2019) applied on top of it for different tasks. All the MTL methods in comparison are trained for 200 epochs, using a batch size of 8. We use Adam as the optimizer with a learning rate of 0.0001 for the first 100 epochs, and with a learning rate of 0.00005 for the rest of the epochs. Following (Liu et al., 2021a) for each

| Method | Segmentation (Higher Better) | | Depth (Lower Better) | | $\Delta m\%\downarrow$ |
|---|---|---|---|---|---|
| | mIoU | Pix Acc | Abs Err | Rel Err | |
| Independent | 74.01 | 93.16 | 0.0125 | 27.77 | - |
| PCGrad (Yu et al., 2020a) | 75.13 | 93.48 | 0.0154 | 42.07 | 18.29 |
| CAGrad (Liu et al., 2021a) | 75.16 | 93.48 | 0.0141 | 37.60 | 11.64 |
| **MoCo (ours)** | 75.42 | 93.55 | 0.0149 | 34.19 | 9.90 |
| **PCGrad + MoCo** | 75.49 | 93.62 | 0.0146 | 46.07 | 20.07 |
| **CAGrad + MoCo** | 75.07 | 93.39 | 0.0137 | 36.78 | 10.12 |

Table 5: MoCo with existing gradient manipulation MTL algorithms for Cityscapes dataset tasks. Results are averaged over 3 independent runs.

| Method | Segmetation (Higher Better) | | Depth (Lower Better) | | Surface Normal Angle Distance (Lower Better) | | Within $t°$ (Higher better) | | | $\Delta m\%\downarrow$ |
|---|---|---|---|---|---|---|---|---|---|---|
| | mIoU | Pix Acc | Abs Err | Rel Err | Mean | Median | 11.25 | 22.5 | 30 | |
| Independent | 38.30 | 63.76 | 0.6754 | 0.2780 | 25.01 | 19.21 | 30.14 | 57.20 | 69.15 | - |
| PCGrad (Yu et al., 2020a) | 38.06 | 64.64 | 0.5550 | 0.2325 | 27.41 | 22.80 | 23.86 | 49.83 | 63.14 | 3.97 |
| CAGrad (Liu et al., 2021a) | 39.79 | 65.49 | 0.5486 | 0.2250 | 26.31 | 21.58 | 25.61 | 52.36 | 65.58 | 0.20 |
| **MoCo (ours)** | 40.30 | 66.07 | 0.5575 | 0.2135 | 26.67 | 21.83 | 25.61 | 51.78 | 64.85 | 0.16 |
| **PCGrad + MoCo** | 38.80 | 65.02 | 0.5492 | 0.2326 | 27.39 | 22.75 | 23.64 | 49.89 | 63.21 | 3.62 |
| **CAGrad + MoCo** | 39.58 | 65.49 | 0.5535 | 0.2292 | 25.97 | 20.86 | 26.84 | 53.79 | 66.65 | -0.97 |

Table 6: MoCo with existing gradient manipulation MTL algorithms for NYU-v2 dataset tasks. Results are averaged over 3 independent runs.

method in comparison we report the average test performance of the model over last 10 epochs, averaged over 3 seeds. For implementing MoCo in Cityscapes dataset we use $\beta_k = 0.05/k^{0.5}$, $\gamma_k = 0.1/k^{0.5}$, where $k$ is the iteration number. For the projection to simplex in the $\lambda_k$ update, we use a softmax function. The training and test loss curves for semantic segmentation and depth estimation are shown in Figures 4a and 4b respectively.

**NYU-v2 dataset.** For NYU-v2 dataset, we follow the same setup as Cityscapes dataset, except with a batch size of 2. For implementing MoCo in NYU-v2 experiments, we use $\beta_k = 0.99$, $\gamma_k = 0.1$ with gradient normalization followed by weighting each gradient with corresponding task loss. This normalization was applied to avoid biasing towards one task, as can be seen is the case for MGDA. For the projection to simplex in the $\lambda_k$ update in MoCo, we apply softmax function to the update, to improve computational efficiency. The training and test loss curves for semantic segmentation, depth estimation, and surface normal estimation for NYU-v2 dataset are shown in Figures 5a, 5b, and 5c respectively. It can be seen that the model start to slightly overfit to the training data set with respect to the semantic loss after the 100th epoch. However this did not significantly harm the test performance in terms of accuracy compared to the other algorithms.

**MoCo with existing MTL algorithms** We apply the gradient correction introduced in MoCo on top of existing MTL algorithms to further improve the performance. Specifically, we apply the gradient correction of MoCo for PCGrad and CAGrad on Cityscapes and NYU-v2. For the gradient correction (update step (6)) in PCGrad we use $\beta_k = 0.99$ for both Cityscapes and NYU-v2 datasets, and for that in CAGrad we use $\beta_k = 0.99$ for Cityscapes dataset and $\beta_k = 0.99/k^{0.5}$ for NYU-v2 dataset. The results are shown in Tables 5 and 6 for Cityscapes and NYU-v2 datasets, respectively. We restate the results for independent task performance and original MTL algorithm performance for reference. It can be seen that the gradient correction improves the performance of the algorithm which only use stochastic gradients. For PCGrad where no explicit convex combination coefficient computation for gradients is involved, there is an improvement of $\Delta m\%$ for NYU-v2 by $0.35\%$. For Cityscapes, it can be seen a slight degradation in terms of $\Delta m\%$, in exchange for improvement in 3 out of 4 performance metrics. This can be expected as PCGrad does not explicitly control the

| Method | Domain | | | $\Delta m\%$ |
|---|---|---|---|---|
| | Amazon | DSLR | Webcam | |
| Mean | 84.22 | 94.81 | 97.04 | - |
| MGDA | 79.60 | 96.45 | 97.96 | 0.94 |
| PCGrad | 84.10 | 95.08 | 96.30 | 0.21 |
| GradDrop | **84.73** | 96.17 | 96.85 | -0.61 |
| CAGrad | 84.22 | 94.26 | 97.41 | 0.07 |
| **MoCo (ours)** | 84.33 | **97.54** | **98.33** | **-1.45** |

Table 7: Results on Office-31 dataset. We show the 31-class classification results over 3 domains on Office-31 data set. The results are obtained from the epoch with the best validation performance.

| Method | Domain | | | | $\Delta m\%$ |
|---|---|---|---|---|---|
| | Art | Clipart | Product | Real-World | |
| Mean | **63.88** | 77.90 | 89.55 | 79.39 | - |
| MGDA | 63.63 | 73.78 | 90.18 | **79.82** | 1.11 |
| PCGrad | 63.06 | 77.46 | 89.09 | 78.70 | 0.81 |
| GradDrop | 63.82 | 78.22 | 89.19 | 78.88 | 0.18 |
| CAGrad | 63.06 | 77.03 | 89.62 | 79.53 | 0.54 |
| **MoCo (ours)** | 63.38 | **79.41** | **90.25** | 78.70 | **-0.27** |

Table 8: Multi-task learning results on Office-Home dataset. We show the 65-class classification results over 4 domains on Office-Home data set. For each method the results are obtained from the epoch with the best validation performance.

converging point to be closer to the average loss. For CAGrad which explicitly computes dynamic convex combination coefficients for gradients using stochastic gradients such that it converges closer to a point that perform well in terms of average task loss, there is an improvement of $\Delta m\%$ for Cityscapes by $1.52\%$ and that for NYU-v2 by $1.17\%$. This suggests that incorporating the gradient correction of MoCo in existing gradient based MTL algorithms also boosts their performance.

In addition to the experiments described above, we demonstrate the performance of MoCo in comparison with other MTL algorithms using Office-31 (Saenko et al., 2010) and Office-home(Venkateswara et al., 2017) datasets. Both of these datasets consist of images of several classes belonging to different domains. We use the method "Mean" as the baseline for $\Delta m\%$, instead of independent task performance. The Mean baseline is the method where the average of task losses on each domain is used as the single objective optimization problem. For reporting per domain performance for all the methods compared in Office-13 and Office-home experiments, the test performance at the epoch with highest average validation accuracy (across domains) is used for each independent run. This performance measure is then averaged over three independent runs.

**Office-31 dataset.** The dataset consists of three classification tasks on 4,110 images collected from three domains: Amazon, DSLR, and Webcam, where each domain has 31 object categories. For implementing experiments using Office-home and Office-31 datasets, we use the experiment setup and implementation given by LibMTL framework (Lin & Zhang, 2022). The MTL algorithms are implemented using hard parameter sharing architecture, with ResNet18 backbone. As per the implementation in (Lin & Zhang, 2022), 60% of the total dataset is used for training, 20% for validation, and the rest 20% for testing. All methods in comparison are run for 100 epochs. For MoCo implementation, we use $\beta_k = 0.5/k^{0.5}$ and $\gamma_k = 0.1/k^{0.5}$. We report the test performance of best performing model based on validation accuracy after each epoch, averaged over 3 seeds. The results are given in Table 7. It can be seen that MoCo significantly outperforms other methods in most taks, and also in terms of $\Delta m\%$.

**Office-home dataset.** This dataset consists of four classification tasks over 15,500 labeled images on four domains; Art: paintings, sketches and/or artistic depictions, Clipart: clipart images, Product: images without background and Real-World: regular images captured with a camera. Each domain has 65 object categories. We follow the same experiment setup as Office-31, and for MoCo imple-

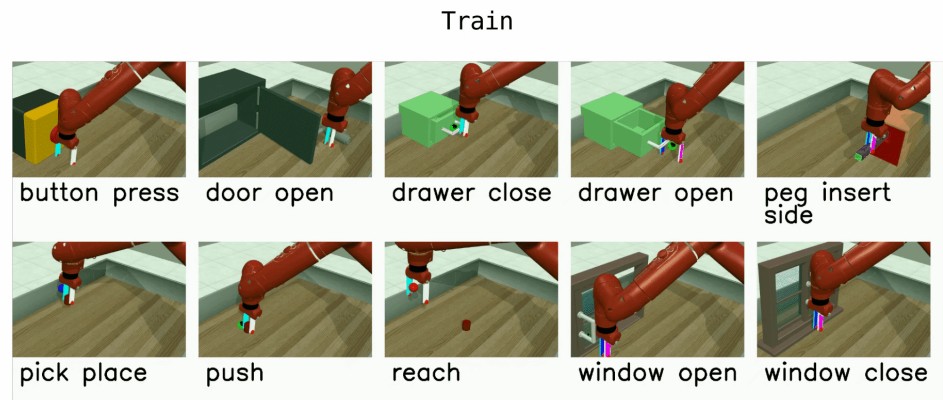

Figure 6: Metaworld MT10 benchmark tasks (Yu et al., 2020b).

mentation, we use $\beta_k = 0.5/k^{0.5}$ and $\gamma_k = 0.1/k^{0.5}$. The results are given in Table 8. It can be seen that MoCo significantly outperforms other methods in most taks, and also in terms of $\Delta m\%$.

| Method | success (mean $\pm$ stderr) |
|---|---|
| Multi-task SAC | $0.49 \pm 0.073$ |
| Multi-task SAC + Task Encoder | $0.54 \pm 0.047$ |
| Multi-headed SAC | $0.61 \pm 0.036$ |
| PCGrad | $0.72 \pm 0.022$ |
| CAGrad | $0.83 \pm 0.045$ |
| **MoCo (ours)** | $0.75 \pm 0.050$ |
| **CAGrad + MoCo (ours)** | $0.86 \pm 0.022$ |
| One SAC agent per task (upper bound) | $0.90 \pm 0.032$ |

Table 9: Multi-task reinforcement learning results on MT10 Metworld benchmark.

## K.3 REINFORCEMENT LEARNING

For the multi-task reinforcement learning setting, we use the multi-task reinforcement learning benchmark MT10 available in Met-world environment (Yu et al., 2020b). Figure 6 illustrates the 10 tasks associated with MT10 benchmark. We follow the experimental setup used in (Liu et al., 2021a) and provide the empirical comparison between our MoCo method and the existing baselines. Specifically, we use MTRL codebase (Sodhani & Zhang, 2021) and use soft actor-critic (SAC) (Haarnoja et al., 2018) as the underlying reinforcement learning algorithm. All the methods are trained for 2 million steps with a batch size of 1280. Each method is evaluated once every 10000 steps and the highest average test performance of a method over 5 random seeds over the entire training stage is reported in Table 9. In this experiment, the vanilla MoCo outperforms PCGrad, but its performance is not as good as CAGrad that optimizes the average performance of all tasks. We further run the gradient correction of MoCo on top of CAGrad, and the resultant algorithm outperforms the vanilla CAGrad. This suggests that incorporating the gradient correction of MoCo in existing gradient based MTL algorithms also boosts their performance.

Table 10 summarizes the hyper-parameters choices used for MoCo in each of the experiments.

| | Cityscapes | NYU-v2 | Office-31 | Office-home | MT10 |
|---|---|---|---|---|---|
| optimizer of $x_k$ | Adam | Adam | Adam | Adam | Adam |
| $x_k$ stepsize ($\alpha_k$) | 0.0001 | 0.0001 | 0.0001 | 0.0001 | 0.0003 |
| $Y_k$ stepsize ($\beta_k$) | $0.05/k^{0.5}$ | 0.99 | $0.5/k^{0.5}$ | $0.5/k^{0.5}$ | 0.99 |
| $\lambda_k$ stepsize ($\gamma_k$) | $0.1/k^{0.5}$ | 0.1 | $0.1/k^{0.5}$ | $0.1/k^{0.5}$ | 10 |
| batch size | 8 | 2 | 64 | 64 | 1280 |
| training epochs | 200 | 200 | 100 | 100 | $2\times10^6$ steps |

Table 10: Summary of hyper-parameter choices for MoCo in each experiment

