# OpenReview forum: "Mitigating Gradient Bias in Multi-objective Learning: A Provably Convergent Approach"
_ICLR.cc/2023/Conference — ICLR 2023 notable top 5%_

### Official Review · Reviewer_Hp6r · 2022-10-24

**Confidence:** 3
**Correctness:** 3
**Technical Novelty And Significance:** 4
**Empirical Novelty And Significance:** 3
**Recommendation:** 8

**Clarity, Quality, Novelty And Reproducibility:**

The paper is well-written; the result is novel to my knowledge, reproducbility is unclear to me.

**Strength And Weaknesses:**

Strengths:
- The paper is well-motivated and well-written, the story line and analysis is easy to follow for experienced readers;
- The MoCo algorithm proposed in this work is the fisrt stochastic multi-gradient method that can converge to Pareto stationary point under mild condition, which is a decent contribution to the field.

Weakness:
- I did not find any obvious weakness of the paper. The notation is a little bit heavy, maybe it is better to add a notation subsection.

**Summary Of The Paper:**

The author(s) studied the multi-objective optmization problem and proposed a new algorithm called MoCo that provably converges to a Pareto stationary point in the stochastic gradient setting. Experiments on both toy and real data are conducted to support the proposed method.

**Summary Of The Review:**

Overall, this paper makes a decent contribution to the field; the author(s) give the first stochastic multi-gradient method can converges under mild conditions. I did not carefully check the proofs, the paper is above the bar of acceptence if other reviewers all agree that the proof is correct.

---

### Official Review · Reviewer_qnbb · 2022-10-24

**Confidence:** 4
**Correctness:** 3
**Technical Novelty And Significance:** 2
**Empirical Novelty And Significance:** 2
**Recommendation:** 8

**Clarity, Quality, Novelty And Reproducibility:**

Clarity: good
Quality: Neutral
Novelty: good
Reproducibility: N/A

**Strength And Weaknesses:**

Strength:
1. The problem studied is important.
2. The writing is easy to follow.

Weakness:
1. It is not clear why multi-objective optimization problems should be formulated as bilevel optimization problems. This critical point should be discussed clearly. At present, it is a weird combination of those two problems.
2. The theoretical analysis is mainly adapted from the bilevel optimization literature. This makes the contribution incremental. The authors should highlight the challenges and the different parts compared with existing bilevel optimization literature.
3. This paper assumes that the loss function is Lipschitz continuous. With this strong assumption, existing bilevel optimization literature can achieve the convergence rate $O(K^{-1/2})$, which is much stronger than the convergence rate in this paper.
4. In lemma 2, the number of objectives $M$ should depend on the number of iterations. This is definitely NOT feasible in real-world applications.

**Summary Of The Paper:**

This paper studied the multi-objective optimization problem. Here, it reformulated it as a bilevel optimization problem and then leveraged the tools in bilevel optimization to establish the convergence rate. Since the main proof is adapted from bilevel optimization, the novelty is incremental.

**Summary Of The Review:**

This paper studied the multi-objective optimization problem. Here, it reformulated it as a bilevel optimization problem and then leveraged the tools in bilevel optimization to establish the convergence rate. Since the main proof is adapted from bilevel optimization, the novelty is good.

---

### Official Review · Reviewer_49fc · 2022-10-25

**Confidence:** 4
**Correctness:** 3
**Technical Novelty And Significance:** 2
**Empirical Novelty And Significance:** 3
**Recommendation:** 8

**Clarity, Quality, Novelty And Reproducibility:**

Clarity: good
Quality: unsure due to the potential issue
Novelty: unsure due to the potential issue

**Strength And Weaknesses:**

Strength:
This is an open and important problem in MOO.

Questions:

I am *pretty* unsure about the correctness of Lemma 4 which indicates the Lipschitz continuity of the lambda. Dontchev & Rockafellar, 2009, Theorem 2F7 does not imply this Lemma.

Using the notation in Dontchev & Rockafellar, 2009, Theorem 2F7, that result holds locally (given $x$, we have a $\mu$) but when we approach the Pareto stationary point ($x$ changes), that $\mu$ might change and we cannot have a global bound of the $\mu$. In other words, the Lipschitz constant of $\lambda$ should actually increase (to infinite) when we are closer to the stationary.

The lambda can be viewed as a solution map of a dual problem of a quadratic programming problem and its Lipschitzness is a very classic problem. To my best knowledge, all of the existing estimations of the Lipschitz are local, not global. Some literatures [1,2,3]


[1] On the continuity of the minimum in Parametric quadratic programs.
[2] On the Lipschitz Behavior of Optimal Solutions in Parametric Problems of Quadratic Optimization and Linear Complementarity
[3] Sufficient Conditions for the Lipschitz Continuity of QP-based Multi-Objective Control of Humanoid Robots

**Summary Of The Paper:**

This paper studies how to mitigate the gradient bias of multiobjective learning. They use the idea of momentum-based gradient bias correction to substitute the gradient when finding the common gradient.

I have had some earlier attempts on this problem but didn't end up very successful and I am very excited that someone also works on this problem. However, I find some potential mistakes in a very core Lemma (Lemma 4) used in this paper and thus I doubt the correctness of the result in this paper.

**Summary Of The Review:**

My main concern is the correct of a core Lemma. My current rating is mainly due to that issue if this can be addressed during the rebuttal period, I am happy to raise the score to 6 or 8. If it can't be addressed, this paper is not ready to publish.

---

### Public Comment · ~Yuchen_Luo3 · 2022-11-19
**Severe Problems  with This Paper**

The article has severe problems in the following aspects, where the correctness of both analysis and experiments does not meet current scores:

1. The result of the convex case (Theorem 1) is wrong, due to the definition where the  $\lambda^*(x_k)$  is adaptive to the solution while the  $x^*$  is defined globally. I have studied this problem for a long time before. I have considered this metric, but found that it has serious problems. Specifically, if we choose  $x^*$  as the optima for  $\lambda^*(x_k) F(x)$, then it could happen that  $\lambda^* (x_j) (F(x_j)−F(x^*))\leq0,j\neq k$ because  $x^*$  is not the optima for $\lambda^*(x_k) F(x)$. Therefore, this metric can not measure the Pareto suboptimality because it may be zero for non-optimal decisions, thus is wrong to analyze Pareto optimality. For example, if $F(x) = (||x-a||\_2\^2,||x-b||\_2\^2)$ where $a=(1,0),b=(-1,0)$, and consider a sequence $x_{2i-1}=(1,1), x_{2i}=(-1,1),i=1,2,\ldots,\frac{K}{2}$, then we have $F(x_{2i-1}) = (1,5),\nabla F(x_{2i-1}) = ((0,2),(4,2))$ and $F(x_{2i-1}) = (5,1),\nabla F(x_{2i-1}) = ((-4,2),(0,2))$. We can calculate that $\lambda^*(x_{2i-1}) = (1,0)$ and $\lambda^*(x_{2i}) = (0,1)$, and next get $\frac{1}{K}\sum_{k=1}^K\lambda^*(x_k) (F(x_k)−F(x^*)) = \frac{1}{K}\sum_{k=1}^K\lambda^*(x_k) (F(x_k)−F(x^*)) = 1-\frac{1}{2}||x^*-a||\_2\^2-\frac{1}{2}||x^*-b||\_2\^2\leq 0$ (the equation is when $x^*=(0,0)$). Note that here the metric is bounded, but the solutions in the sequence are obviously not Pareto optimal. The correct formulation should be $\frac{1}{K}\sum_{k=1}^K\lambda^*(x_k) (F(x_k)−F(x_k^*))$ where $x_k^* = \arg\min_{x} \lambda^*(x_k) F(x)$, or just to bound $\lambda^*(x_k) (F(x_k)−F(x^*)),\forall x^*$ without the average scheme. Current results can analogize to the intermediate result in Lemma 4.2 (average scheme) in [1], but [1] provides complete convergence result in Theorem 4.1. In addition, the average scheme in single-objective stochastic optimization can always be transferred to the traditional version [2,3], which is why the average scheme is meaningful for proving convergence. However, Theorem 1 in this paper can not reduce to the traditional version, which causes the bound for the summation  $\frac{1}{K}\sum_{k=1}^K\lambda^*(x_k) (F(x_k)−F(x^*))$  meaningless to reveal convergence information for solution  $x_k$.

2. I have also followed the version submitted in NeurIPS 2022. The experimental results are exactly the same, but the algorithm and parameters setting have been changed a lot. Therefore, the experimental results are not trustworthy.

3. I have followed this area for a long time, and found that there is a related work [4] in NeurIPS 2022. The convergence results in [4] are aligned with single-objective sgd, but the rate in this paper is much slower. Could the authors provide a more detailed comparison with [4]?

[1] Fliege et al. Complexity of gradient descent for multiobjective optimization

[2] Cutkosky et al. Momentum-based variance reduction in non-convex sgd.

[3] Drori et al. The complexity of finding stationary points with stochastic gradient descent.

[4] https://openreview.net/forum?id=ScwfQ7hdwyP

---

### Author Response · Authors · 2022-11-21
**Response to Comments from "Yuchen Luo" ( II )**

**Q3. Comparison with concurrent work.**

Thanks for sharing this interesting concurrent work. As the paper is not available before ICLR deadline and the revision period has just ended, we can only cite and discuss it in the next version.

We quickly went through [3] over the weekend and found the following differences between MoCo and [3].

- *Difference in algorithms.* The focus of [3] is to introduce a unified framework for stochastic gradient-based MOO algorithms. The similarity between MoCo in this submission and [3] is that both algorithms use momentum-like techniques to reduce the gradient bias. The key difference between two algorithms is that MoCo uses momentum-based  correction on the full gradient estimation while [3] uses the momentum-like moving averaging for $\lambda$. As a consequence of this difference, it is not clear if the averaged weight $\lambda$ obtained in [3] will ensure convergence of the stochastic MOO update direction to any deterministic MOO direction, but in our work, we can show the convergence to MGDA direction in Lemma 1.

- *Difference in theorems.* The difference of the convergence rates is mainly due to different assumptions used in two papers. Theorems 1 and 2 in our paper use fairly standard assumptions in optimization literature. To achieve the scalar-valued SGD rate ${\cal O}(K^{-\frac{1}{2}})$, the theorems in [3] require additional bounded assumption on the function values.  This is precisely the reason why the results for Theorems 1, 2 are not comparable to the results provided in [3]. Requested by Reviewer qnbb, we have also provided the faster convergence rate under the assumption on the bounded function value in Theorems 3 and 4. The rate in Theorem 3 is slower compared to Theorem 1, 2 of [3] due to the weaker assumption of biased gradient used in Theorem 3. Theorem 4 provides the ${\cal O}(K^{-\frac{1}{2}})$ rate matching the standard SGD, with still weaker assumptions on the bias on the stochastic gradient compared to Theorem 1, 2 of [3].

Thanks again for your continuing interest in our paper.

Authors

[1] Liu, Suyun, and Luis Nunes Vicente. "The stochastic multi-gradient algorithm for multi-objective optimization and its application to supervised machine learning." Annals of Operations Research (2021): 1-30.

[2] Introduction to Online Convex Optimization, Foundation and Trends in Optimization, Elad Hazan 2016

[3] https://openreview.net/forum?id=ScwfQ7hdwyP

---

### Author Response · Authors · 2022-11-21
**Response to Comments from "Yuchen Luo" ( I )**

Thanks for your interest in our paper and the MOO area. We provide our response to your comments next. In short, your assertion is not true.

**Q1. Metric used in the convex setting.**

The error metric we use for the convex setting is similar/exchangeable to the error metric used by [1], which is, to the best of our knowledge, the only work that analyses a stochastic counterpart of an existing gradient based MOO algorithm. Here, we will restate the error metric used in [1] as
$$\quad\quad\quad\quad\quad\quad\quad\quad\quad\quad\quad\quad\quad\quad\quad\quad  \min_{k\in[K]} \mathbb{E}[F(x_k)\lambda^*(x_k)] - \mathbb{E}[F(x^*)\bar{\lambda}_K],$$

where $ \bar{\lambda}\_K = \sum_{k=0}^K \lambda^*(x_k) $, and $x^*\in\mathcal{X}$. Note that the error metric we use is an upper bound of the above error metric.

Furthermore, the negative error metric is not uncommon, especially in situations where the objective function is adaptive to the decision $x_k$. By defining a time-varying function $f_k(x_k):=F(x_k)\lambda^*(x_k)$, our performance metric can be understood as the static regret commonly used in online convex optimization literature, which is defined as [3]
$$\quad\quad\quad\quad\quad\quad\quad\quad\quad\quad\quad\quad\quad\quad\quad\quad {\rm Regret}:=\sum_{k=1}^K f_k(x_k)-\min_x \sum_{k=1}^K f_k(x).$$
It is well-known that the static regret can be also negative for some round $k$ and there are improved metric developed afterwards, but this does not disqualify regret as a good performance metric used in thousands of papers.

We note that this is nothing to do with the correctness of our analysis, rather, this is related to what is a good metric to use in this line of research. Of course, developing other non-negative metrics in similar spirit of dynamic regret is of interest, but it is going beyond the scope of our work. We will mention this in the future direction.

**Q2. Experimental results of MoCo.**

We respectfully disagree your assertion. In fact, our experimental results are trustworthy and reproducible.

As we have explicitly mentioned in our submission, this is a resubmission. The key component we have revised is the theoretical part not the algorithm. Specifically, we have removed the unjustifiable assumption and introduce a new regularization technique to control $\lambda^*$, both of which will only affect the choice of constants in the theorems. However, the regularization is crucial only for the convergence to the true MGDA descent direction. We have also added results without the regularization but under different assumptions in Theorems 3 and 4. In simulation sections, we have explicitly listed the choices of stepsizes in each experiment, so our empirical results remain unchanged with the same hyperparameters for this submission. We are providing the source code to the AC and the reviewers for reproducibility.

---

### Decision · Program_Chairs · 2023-01-20

**Decision:**

Accept: notable-top-5%

**Justification For Why Not Higher Score:**

N/A

**Justification For Why Not Lower Score:**

The paper technically sounds and solving an important problem. Moreover, the contributions are both theoretical and empirical. Hence, it is clearly a large interest within the community. I believe there is some significant knowledge for majority of the audience, specifically:
- ML Practitioners: The resulting algorithm is effective in various practical problems including scene understanding and reinforcement learning both are active research sub-fields.
- Theory: The theoretical advancement is non-straightforward and relates to both online learning and optimization theory.
- Multi-objective optimization/Multi-task learning: This is the core subfield where the paper lies. It would have a major impact on this core sub-field.

**Metareview: Summary, Strengths And Weaknesses:**

The paper is proposing an analyzing a stochastic multi-objective optimization method. Existing multi-objective optimizers are typically posed as solving a meta-optimization problem using gradients. However, these gradients are stochastic resulting a bias in the final solution. The proposed method uses an adaptive approach to track optimization solution through time effectively alleviating bias. Method is further fully analyzed in convex and non-convex setting. Moreover, it is evaluated in various interesting applications including supervised learning and RL. Hence, it is a solid work on an important problem. Moreover, three experts reviewed the paper independently and raised various issues which are all addressed by the authors during the discussion phase. In their final scores, all reviewers agree on the quality and acceptance.

**Note From Pc:**

if the above contains the word "oral" or "spotlight" please see: "oral" presentation means -> notable-top-5% and "spotlight" means -> notable-top-25%. As stated in our emails, we are disassociating presentation type from AC recommendations